# Quantitative proteome of bacterial periplasmic predation by *Bdellovibrio bacteriovorus* reveals a prey-lytic protease

Ting F. Lai ®[1], Denis Jankov ®[1], Jonas Grossmann ®[2,3], Bernd Roschitzki ®[2] & Simona G. Huwiler ®[1] ✉

The rise of antimicrobial resistant pathogens calls for novel ways to kill and damage bacteria. A rich source for bacterial cell-damaging proteins is the periplasmic predatory bacterium *Bdellovibrio bacteriovorus*, which invades, kills and subsequently exits the prey cell. An increased understanding of predatory protein function can be achieved by analyzing their relative abundance at key stages of predation. We present the first quantitative proteome covering the complete predatory life cycle of *B. bacteriovorus* killing *Escherichia coli*, quantifying 2195 predator proteins. Protein clustering reveals nine distinct clusters sharing similar expression patterns. Notably, the protease Bd2269 is highly abundant during the prey exit phase. Gene knockout and heterologous expression experiments reveal that Bd2269 is involved in the prey exit process and lyses *E. coli* from within. Our quantitative predator proteome is a valuable resource to study specific stages of the predatory life cycle, contributing to advancing the search for novel antimicrobial enzymes.

The rise and scale of antimicrobial resistance[1] calls for alternative approaches to combat multidrug-resistant (MDR) Gram-negative bacterial pathogens, such as utilizing predatory bacteria[2,3]. Periplasmic predatory bacterium *Bdellovibrio bacteriovorus* is regarded as a 'living antibiotic'[4] due to its low immunogenicity[5] and capability to kill many MDR Gram-negative pathogens, including many on the WHO priority pathogens list[6], irrespective of their antibiotic resistance[7,8]. Further advantages of *B. bacteriovorus* include the ability to reduce biofilms formed by MDR Gram-negative pathogens[7] and to prevent or disrupt biofilms of the Gram-positive *Staphylococcus aureus*[9]. When encountering the latter, it is known to secrete a multitude of proteases[9,10].

*B. bacteriovorus* is a model organism for periplasmic predation, as it is genetically tractable and found in many different environments[11]. This predator invades through the outer membrane and peptidoglycan of the prey and establishes itself in the prey periplasm, feeding on the prey's cytoplasmic contents as a main nutrient source. After the predator has divided, its progeny cells lyse the prey cell remnants to escape and search for new prey[11–13] (Fig. 1a). Understanding this fascinating predatory life cycle is important for a potential future medical application of *B. bacteriovorus*, next to learning about this predator-prey interaction. Approximately one-third of the genes in the *B. bacteriovorus* HD100 genome encode proteins whose

function is unknown and lack recognizable homology to proteins from other (non-predatory) bacteria[14]. As an organism with one of the highest densities of hydrolytic enzymes in the genome[15], a predatory proteome would be ideal to discover novel, unique, or modified enzymatic functions that could serve as a potential source of new antimicrobial drugs.

Bacterial-damaging predatory enzymes are needed when crossing the outer membrane and peptidoglycan of the bacterial prey, which first occurs upon invasion of the prey and again at the end when the predator exits from the prey cell remnants. Many aspects of how the predator escapes and lyses the remaining prey structures are still unknown[16–18]. The genome of *B. bacteriovorus* encodes ~10% hydrolytic enzymes, including 150 predicted protease genes[14], of which a subset could aid the predator to escape the depleted prey. For example, the hydrolytic capability of host-independent *B. bacteriovorus* HIB strain to clear *S. aureus* biofilm involves various proteases and peptidases (e.g., Bd2269, Bd2321, and Bd2692) secreted to its culture supernatant[9]. Subsequently, direct application of purified proteases Bd2269 and Bd2692 to a *S. aureus* biofilm effectively reduced it, likely affecting the biofilm's complex network[9,10,19]. While these represent some examples of known hydrolytic and bacterial-damaging functions, the vast majority of proteins with hydrolytic or unknown function in *B. bacteriovorus* remain uncharacterized. To facilitate this, additional

[1]Department of Plant and Microbial Biology, University of Zurich, Zurich, Switzerland. [2]Functional Genomics Center Zurich, University of Zurich/ETH Zurich, Zurich, Switzerland. [3]Swiss Institute of Bioinformatics (SIB) Quartier Sorge - Batiment Amphipole, Lausanne, Switzerland. ✉e-mail: simona.huwiler@uzh.ch

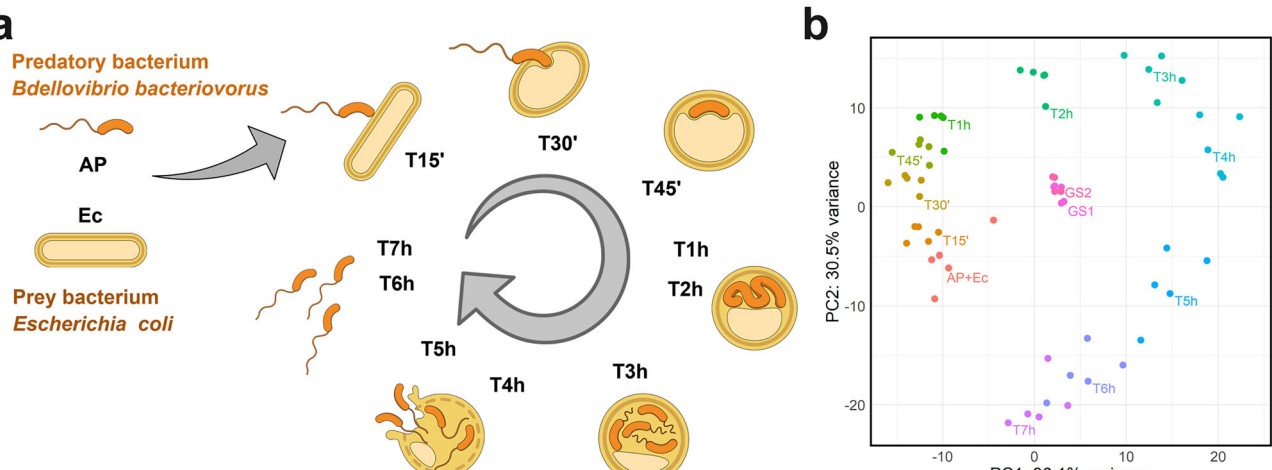

**Fig. 1 | Quantitative proteomics data reflects the cyclic nature of the predatory life cycle. a** Predatory life cycle of *Bdellovibrio bacteriovorus* HD100 with *Escherichia coli* K-12 MG1655 as the prey bacterium. *B. bacteriovorus* starts as a free-swimming attack phase (AP) cell, which attaches onto the prey bacterium, before invading into its periplasm. *B. bacteriovorus* then grows inside the prey cell, utilizing its cell contents for growth, before division into multiple progeny cells and lysing the depleted prey cell to return to AP. The samples taken for proteomic analysis at each timepoint are highlighted in bold (T15'-T7h). **b** Principal component analysis (PCA) of normalized proteomic data reveals samples grouped in a circular pattern mimicking the predatory life cycle stages. The "AP+Ec" condition combines the protein abundances from "attack phase (AP)" and "*E. coli*-only (Ec)," for better comparison with other conditions. "Golden standard (GS)" conditions comprise an equal mixture from every sample from each sample from the predatory life cycle, which allows comparisons between each mass spectrometry TMT experiment. Being a mix of all samples, the golden standards (GS1, GS2) plot to the center of the PCA. Five biological repeats were performed for each condition.

information on protein abundance at the different phases of the predatory life cycle may be helpful.

Holistic, high-throughput genomic and transcriptomic studies on *B. bacteriovorus*[20–22] facilitated research on mechanisms of predator-prey interactions. For example, early transcriptomics studies revealed a "predatosome" of genes involved in invasion[22], aiding research on the initial phase of prey entry[23,24]. mRNA levels have served as a proxy for protein levels, although the correlation between mRNA and protein abundance has generally not been shown to be very strong, e.g., due to additional transcriptional and translational regulation[25–27]. Since proteins carry out cellular functions, using proteomics to directly measure protein expression can provide a more accurate view of the cellular processes.

To date, studies using proteomic approaches in *B. bacteriovorus* have been performed using a label-free shotgun approach[9,24,28,29]. An early study by Dori-Bachash and colleagues identified differentially expressed proteins from attack phase (AP) and growth phases in *B. bacteriovorus* HI-6, a host-independent strain[29]. More recently, Tyson et al.[24] identified that deletion of Bd0875, a predatory MIDAS-family adhesin protein, leads to ~10% of rounded, but uninvaded prey cells. This population was analysed at the end of the predatory life cycle, revealing a list of secreted proteins into the prey that was deemed sufficient for prey killing[24]. Generally, in label-free data-dependent shotgun proteomics, the quantification is based on MS1 measurements, where the intensity of peptide precursor ions is assumed to correlate with protein abundance in the sample. However, relying solely on unique peptides as markers for protein abundance may not necessarily capture differences between samples. Additionally, challenges inherent to label-free proteomics, like retention time variability and missing values, may contribute to inaccurate identification or quantification[30].

Tandem mass tag (TMT)-based isobaric labeling enables precise peptide quantification by measuring the reporter ion intensity of the TMT label of a given peptide. By labeling each sample with different TMT labels and processing them together, a direct ratio can be derived between the different samples based on their distinct reporter ions, allowing precise quantitative comparisons. The ability to multiplex samples and use offline prefractionation further increases the sensitivity and accuracy of proteomics analysis, in contrast to shotgun proteomics. Therefore, TMT-based quantitative proteomics is ideal to study the predatory life cycle of *B. bacteriovorus* and provides a precise snapshot of the protein levels.

In this study, we generated a highly resolved quantitative proteome of predatory proteins identified throughout the predatory life cycle using TMT labeling. This dataset comprises 61% of *B. bacteriovorus* proteins at high confidence that are present over the entire predatory life cycle on *Escherichia coli* K-12 MG1655. We provide the protein abundances throughout the predatory life cycle and $\log_2$ of the fold changes [$\log_2$(FC)] of each protein compared to the free-swimming AP to inform hypothesis generation on predatory proteins of unknown function. Further, we identified nine distinct protein clusters, exhibiting similar expression patterns. As one example, we show that one of the most highly abundant proteases in the exit phase, Bd2269, is involved in escape from the depleted prey cell and caused *E. coli* cell lysis upon heterologous expression. Beyond proteases, this highly resolved quantitative predatory proteome offers a useful resource to investigate a wide range of biological processes underlying this bacterial predation. Ultimately, we show that it helps to identify cell-damaging predatory enzymes potentially useful to combat MDR pathogens.

## Results

### Quantitative predator proteome analysis of 2195 proteins over the whole predatory life cycle

To support hypothesis generation of predator enzymes of unknown function and gain a holistic overview on the cell biology of periplasmic predator *B. bacteriovorus*, we generated a quantitative proteome over the whole predatory life cycle. Samples collected at various time points of the predatory life cycle, ranging from 15 min to 7 h after mixing predator with the model organism *Escherichia coli* K-12 MG1655 (a laboratory strain) as prey (Fig. 1a, T15' - T7h), were processed. Using TMT isobaric labeling we generated a high-quality predator proteome covering all phases of the periplasmic predatory life cycle from five biological and independent repeats (Fig. 1a). After normalization of protein abundance (Supplementary Fig. 1), 2195 predator proteins were quantified throughout the full predatory life cycle (Supplementary Data 1). This corresponds to approximately two-thirds of all *B. bacteriovorus* HD100 predicted proteins[14]. Although small proteins were underrepresented due to the inherent limitations of TMT-based proteomics, membrane proteins were not biased against (Supplementary Fig. 2). To assess the quality of the quantitative dataset, various analyses were conducted.

## High data quality reflecting cyclic nature of predatory life cycle confirmed by principal component analysis

Principal component analysis (PCA) of the quantitative predatory proteome effectively clusters samples from each timepoint of the predatory life cycle, mirroring the different phases without batch effects (Fig. 1a, b). The circular pattern of the samples in the PCA, arranged in temporal order, reflects the cyclic nature of the life cycle. The 7-h samples gravitate back to the AP starting condition, indicating that the protein abundance levels of new predator progeny are again similar to the initial AP. A hierarchically clustered heatmap effectively groups the phases of the predatory life cycle (Supplementary Fig. 3). Samples of the attack and invasion phase, growth (T2h), division (T3h and T4h), as well as late exit and reversion to AP following exiting (T5h, T6h, T7h) are grouped similarly, further validating the quality of the data. Both the PCA and hierarchically clustered heatmap confirm the high reproducibility and quality of the normalized data across the different predatory life cycle phases. The resulting quantitative data (Supplementary Data 1) serve as a valuable resource to track predatory protein abundance over the whole predatory life cycle for hypothesis generation.

## Identification of nine distinct clusters of predatory proteins with similar expression trends throughout the predatory life cycle

To facilitate hypothesis generation for the function of unknown predatory proteins, proteins with similar expression patterns were clustered. Protein abundances were normalized to "AP+Ec" so all proteins start at a value of 0. The hierarchically clustered heatmap shows protein expression changes over time (Fig. 2a). Based on hierarchical clustering, the data were initially divided into 18 clusters. Due to similar expression patterns of some clusters, these 18 clusters were manually grouped into nine distinct clusters (Fig. 2b and Supplementary Data 2). These nine clusters reflect specific expression patterns distinct for different phases of the life cycle.

A functional cluster of orthologous genes analysis of these protein clusters revealed that up to half of the proteins in some clusters have unknown functions (Fig. 2c). In clusters A and B, where protein abundance increases during prey cell entry and bdelloplast establishment, the highest functional category, aside from unknown, is cell wall/membrane and envelope biogenesis. This likely reflects the predatory proteins involved in manipulating and remodeling of the prey cell upon entry, many of which have been described in connection with prey peptidoglycan manipulation[23,31,32]. In cluster D, peaking at 3h with slow drop towards the end of the periplasmic predation cycle, are mainly the categories of transport and metabolism of inorganic ions, carbohydrate, and coenzymes, as well as replication, recombination, and repair, consistent with periplasmic feeding and growth during this phase. In clusters E and G, which peak at 4 h, a major functional category is cell cycle control, cell division, and chromosome partitioning, expected with progeny cell division at that phase. Finally, clusters G and I, increasing from 3 h onward, show a large proportion of cell motility proteins, indicating flagellar synthesis for predator progeny cells. Additionally, cluster I also shows increased signal transduction, indicating an increased need for stimulus integration in new predatory cells. Although analyzing protein expression trends provide insights into proteins that may have similar functions or interactions, performing pairwise comparison between two different phases of the predatory life cycle by differential expression enables a detailed view of the regulated proteins aiding in functional assignments.

## Pairwise comparisons of each predatory life cycle phase with initial attack phase

To enable a detailed overview of protein expression at a specific timepoint during the predatory life cycle, pairwise comparisons were performed at each timepoint to the "AP+Ec" condition at the beginning of the experiment. Detailed lists of proteins with their $\log_2$(FC) and false discovery rate (FDR) for each timepoint are provided in Supplementary Data 3. To improve the current *B. bacteriovorus* HD100 genome annotations (NCBI Taxonomy ID: 264462), functional annotations from eggNOG-mapper v2.1.12[33], and the eggNOG e-value were integrated into Supplementary Data 3 to improve hypothesis

generation for hypothetical proteins compared to the original one[14]. Volcano plots showing $\log_2$(FC) and $-\log_{10}$(FDR) for each protein and timepoint provide an overview, highlighting significantly regulated proteins (Fig. 3). As significance threshold on the FDR (adjusted p-value according to Benjamini–Hochberg) was set at 0.05. Up- or down regulation was considered for 2-fold increase or decrease. Different proteins described in literature are highlighted and their abundance discussed in the following paragraphs.

## Protein changes in free-swimming predator form

*B. bacteriovorus* in its free-swimming AP requires a flagellum. We detect an increase in protein abundance of flagellar proteins from T4h to T7h in cluster I (Fig. 2b and Supplementary Data 2), namely flagellar basal-body proteins FlgB and FlgF (Bd3407, Bd0530), and flagellin homologs FliC (Bd0604, Bd0606). As expected during this time, new flagella are formed in the predator progeny cells. Additionally, a probable chemoreceptor glutamine deaminase CheD (Bd2829) is found in this cluster, indicative of an increased need for chemotaxis in AP (Cluster I, Fig. 2b). Comparing AP cells at T6h and T7h to the older 24-hr lysate AP cells may reveal what adaptations new progeny cells undergo (Fig. 3 and Supplementary Data 3). In these newly formed progeny cells (T6h and T7h), enzymes involved in reducing oxidative stress—such as glutathione peroxidase (Bd0326), superoxide dismutase (Bd1401), and glutaredoxin (Bd2887)— are upregulated by more than 2-fold (vs. "AP+Ec"). The three most upregulated proteins of AP at T7h (vs. "AP+Ec") are three proteins of unknown function (Bd1524, Bd2244, Bd3489). Furthermore, Bd0119 (Flp1), a pilin known to be genetically expressed in AP[34,35], is upregulated ~2.7-fold in the progeny AP cells at T6h and T7h (vs. "AP+Ec", Fig. 3, Supplementary Data 3). This pilin abundance increased markedly from T4h to T5h, likely representing the start of pilin assembly in progeny AP cells (Supplementary Data 1).

## Proteins involved in prey attachment, entry, and periplasmic niche establishment

Upon attachment to the prey (T15'), the two most strikingly upregulated proteins compared to AP have unknown functions (7.3-fold for Bd0434, 6.2-fold for Bd3155) (Fig. 3 and Supplementary Data 3). While Bd0434 encodes a secreted α-helix of unknown function, Bd3155 is a predicted C-type lysozyme inhibitor (UniProt prediction[36,37]). However, protein docking analysis between this lysozyme inhibitor and predator lysozyme Bd1411 revealed no evidence of coevolution/co-folding between these two proteins. Further, Bd0434 and Bd3155 remain amongst the most upregulated proteins during the whole entry phase into the prey (T30'-T1h). During prey entry (T30'), flagellins Bd0604 and Bd0606 are downregulated more than 2-fold (vs. "AP+Ec") consistent with the observed flagellar retraction[38] and its likely degradation. Additionally, we found four proteins which are substantially downregulated during the phases of periplasmic predation: two uncharacterized proteins (Bd1815, Bd2577), regulatory protein (Bd3071), as well as an amidophosphoribosyltransferase (Bd3009) (Fig. 3 and Supplementary Data 3).

Later, during the invasion to early bdelloplast phases (T30'-T1h), many prey cell wall-modifying enzymes are upregulated (over 2-fold). In addition to the DD-endopeptidases Bd0816 and Bd3459, which are responsible for changing the rod-shaped prey into a spherical bdelloplast[23], the L-D-transpeptidases Bd0886 and Bd1176 modify the prey peptidoglycan cell wall[32]. From the remaining 17 L-D-transpeptidases that *B. bacteriovorus* HD100 encodes[32], Bd3176 and Bd3741 were also found highly expressed at this stage (Fig. 3 and Supplementary Data 3). N-acetyl glucosamine deacetylases (Bd0468, Bd3279), which are known to modify the prey peptidoglycan for destruction at the exit phase[31,39], are also upregulated. These modifications highlight the importance of the bdelloplast niche formation for the predator's use, validating our approach.

Interestingly, many proteins of a gliding motility cluster (Bd0411-Bd0421) are highly upregulated at the early bdelloplast phase (T45' and T1h, Fig. 3 and Supplementary Data 3). All proteins identified in this gliding cluster have a peak protein abundance at T45' or T1h (Supplementary

**Fig. 2 | Classification of *B. bacteriovorus* proteins identified by quantitative proteomics into clusters based on their protein abundance throughout the predatory life cycle.** Protein abundances are normalized to the "AP+Ec" condition (starting at 0 h, "attack phase (AP)" and "*E. coli*-only (Ec)"). **a** Heatmap with classification of proteins of similar expression patterns throughout the predatory life cycle into 18 clusters. High expression proteins are in blue, low expression in red (units are scaled normalized protein abundance). The *y*-axis shows a hierarchical tree with 18 clusters (colored bars). The *x*-axis presents each proteomic sample timepoint chronologically, with each row representing a distinct protein. **b** Graphical representation of nine protein clusters showing expression patterns over the predatory life cycle. The 18 clusters from **a** were merged into nine based on expression similarities. "#" depicts the number of proteins within each cluster. The protein identifiers for each cluster can be found in Supplementary Data 2. **c** Functional cluster of orthologous genes (COG) analysis on each protein cluster in **b** assigning predicted functions. Annotations are based on COGs from eggNOG-mapper v2[33].

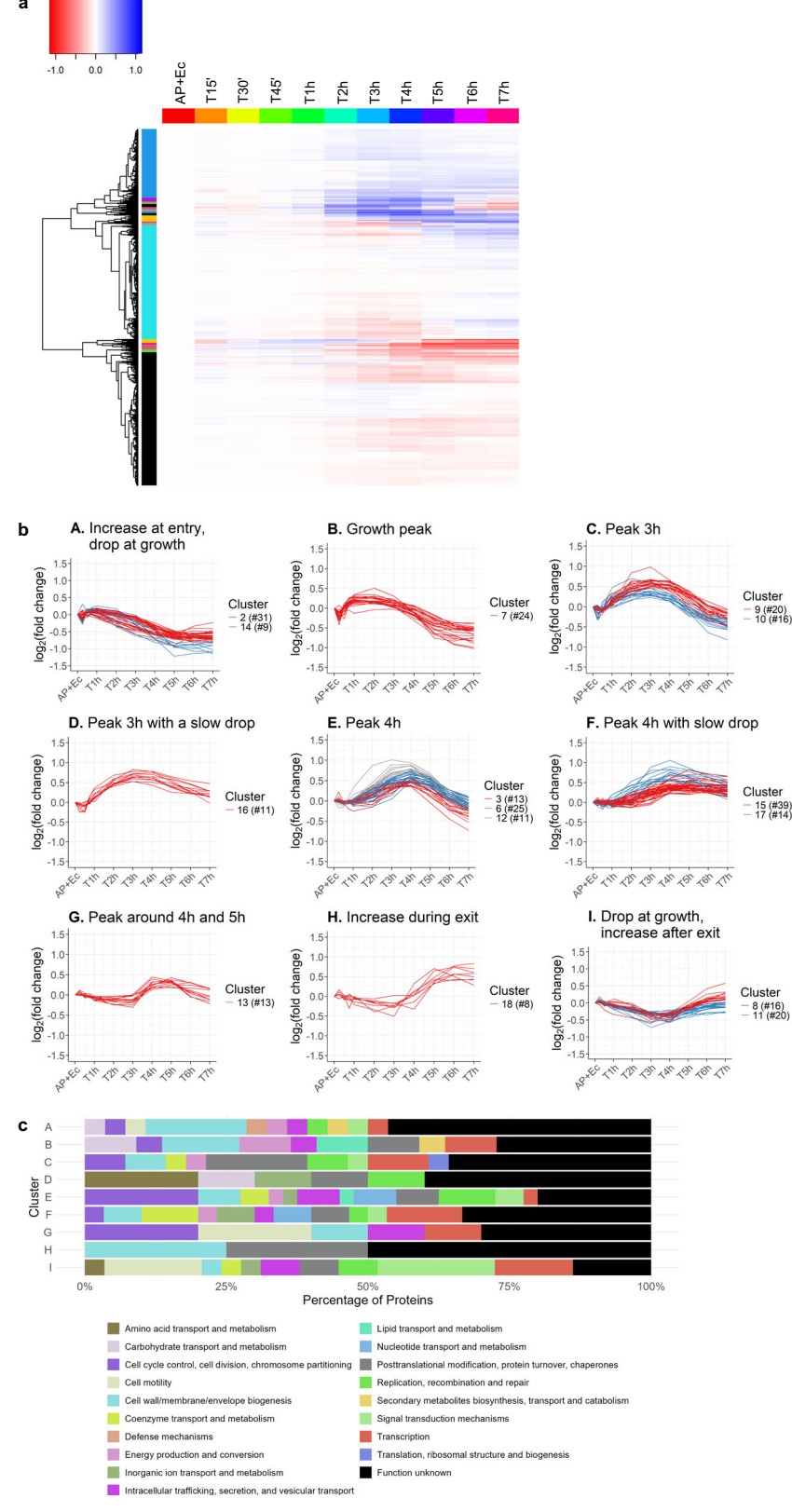

Data 1). As most predator cells would have entered the prey periplasm at these timepoints, we speculate that this gliding cluster is used to move inside of the bdelloplast to establish itself within its new niche. Overall, our quantitative proteomics dataset is in good agreement with previously published findings of prey invasion, while allowing for additional insights.

## Upregulated predator proteins during growth within prey periplasm and division into progeny cells

The two most upregulated proteins during the growth phase at T2h (vs. "AP +Ec") are uncharacterized proteins Bd0278 and Bd2997. Interestingly, operon Bd0276-Bd0284 has been previously implicated in fatty acid

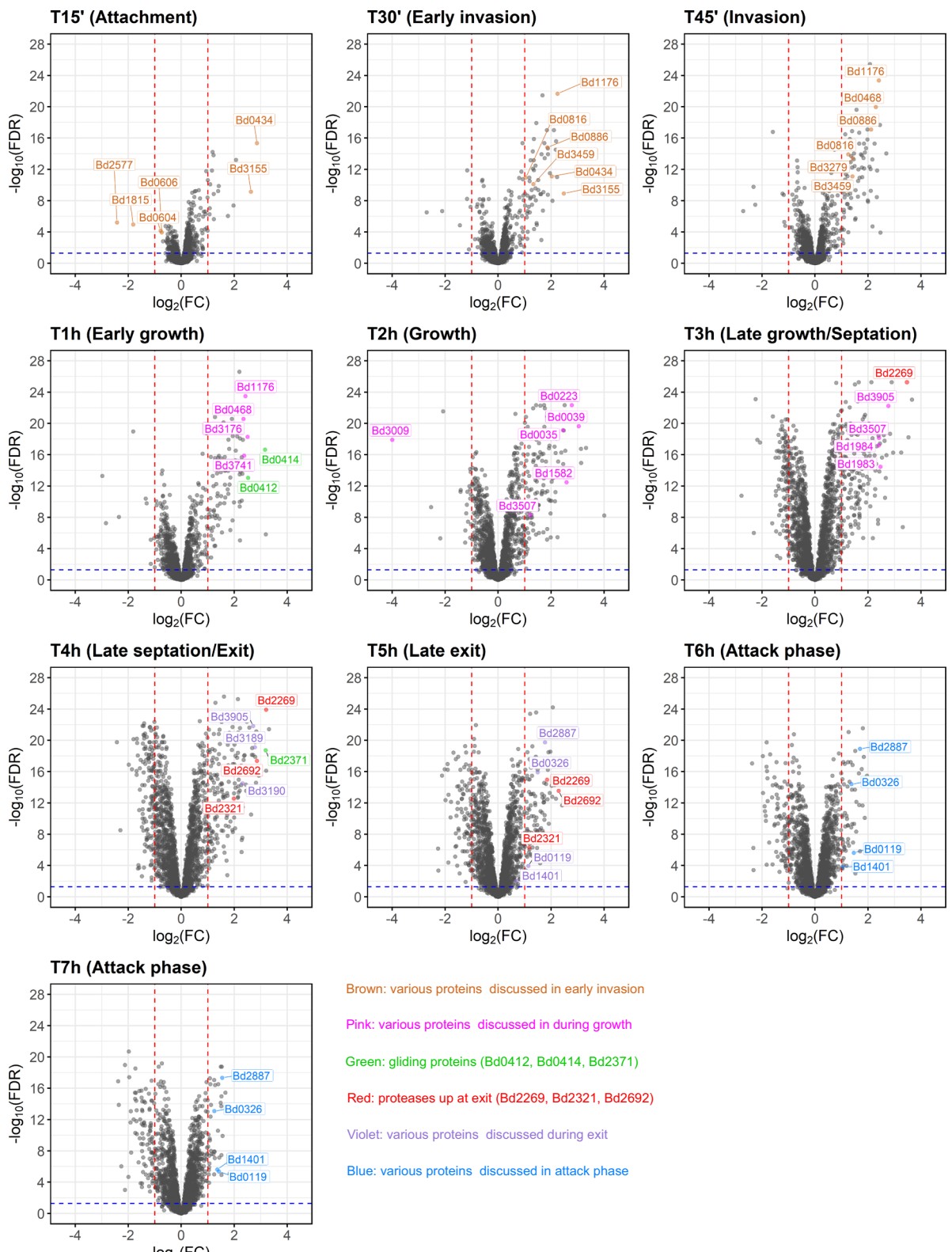

**Fig. 3 | Volcano plots comparing the log₂ (fold change) of predatory proteins at timepoints throughout the predatory life cycle compared to attack phase.** Each panel highlights proteins previously identified in literature to compare to the quantitative proteomics dataset, and highly up- or downregulated proteins which may potentially contribute to an important function within the predatory life cycle. Proteases that are tested in this study are shown in red. Proteins belonging to putative gliding motility clusters[22] are shown in green. A log₂ (fold change, FC) of 1 (2-fold increase) and -1 (2-fold decrease) in protein abundance are indicated by red dotted lines. A -log₁₀(FDR refers to the adjusted *p*-value, FDR) of 1.3 is marked by a blue dotted line (equal to an FDR of 0.05). A detailed list of the compared data, including eggNOG-mapper v2 annotations[33] is in Supplementary Data 3.

synthesis[40], which aligns with the metabolic context of the growth phase. In addition, several other proteins are highly upregulated (>4-fold), including Bd0039 (DNA inversion stimulation factor), Bd0223 (granula-associated regulation protein), Bd0512 (RecA), Bd1582 (single stranded DNA-binding protein), Bd0035 (trypsin-like serine protease), Bd3905 (chromosome partitioning protein, ParB[41]), and Bd0575 (putative imelysin). Most of these proteins are growth related and involved in crucial processes like DNA replication and division. We speculate that Bd0223 may act as a storage for the uptake of prey nutrients that might not be used immediately. On the other hand, Bd3009 (amidophosphoribosyltransferase) is heavily down-regulated (FC = 0.06 at T2h) throughout the predatory life cycle, starting from invasion at T30'. As Bd3009 is predicted to be used in the de novo synthesis of purines[42], it could enable *B. bacteriovorus* to synthesize its own purines in the absence of prey, but is suppressed when extracting nucleotides, including purines, from the prey. Supporting this hypothesis is a secreted nuclease Bd3507 (FC = 5.35, 5.17 at T3h, T4h, respectively), which is upregulated during the late bdelloplast phase, which could degrade prey DNA to release nucleotides for uptake and integration into the predator DNA.

A pair of ribonucleoside reductase proteins (Bd1983, Bd1984) was found to be highly upregulated from T2h to T3h (FC from 2.03 to 5.54; 1.87 to 5.32, respectively). These proteins are implicated in DNA synthesis[43] by converting ribonucleotides into deoxyribonucleotides, which indicates that *B. bacteriovorus* redirects the RNA building blocks to DNA synthesis at T3h. The shift towards intense DNA replication is reflected by RecA (Bd0512), which peaks in abundance at T3h and T4h, being upregulated throughout periplasmic predation with FC > 2 for T2h-T5h (Fig. 3, Supplementary Data 1 and 3, Table 1). During septation (T3h-T4h), we detected cell division proteins FtsZ and FtsA (Bd3189, Bd3190) among the highest upregulated proteins (Table 1 and Supplementary Data 3). Further, proteins of the ParABS system (Bd3906—ParA, Bd3905—ParB), which contribute to chromosomal segregation during cell division[41] are also highly upregulated. Interestingly, DivIVA (Bd0464), which controls cell morphology of new progeny cells[44] becomes only highly expressed at T4h (FC = 2.6) but remains low at T3h (FC = 0.78), aligning with new progeny formation.

### Proteases Bd2269, Bd2692, and gliding motility are prevalent in the exit phase

Our quantitative proteomics dataset at the exit phase (T4h) uncovers proteins highly upregulated compared to attack phase (4-fold upregulation, Table 1). Proteins that were already highly upregulated during late growth and septation (T3h), including those involved in DNA synthesis, replication, division, and flagellar assembly (Bd0543), remain upregulated compared to AP, indicating ongoing division and assembly of new predator progeny cells. Examining the dataset for potential prey cell-damaging enzymes revealed two serine proteases Bd2269 and Bd2692 among the most upregulated proteins at the exit phase. We hypothesize that these proteases, which are predicted to be secreted[45], degrade proteins in the periplasm of the bdelloplast to facilitate escape. Furthermore, many proteins from gliding cluster *bd2368-bd2377* were highly upregulated at the exit phase (Table 1), suggesting that gliding motility is used in the escape of the progeny cells from the prey remnants. While this result reveals which of the four complete gliding clusters[46] in *B. bacteriovorus* HD100 is involved in the exit process, even in liquid culture, gliding motility was previously shown to be involved in the exit phase, regulated by Bd0367[16,18]. Finally, ~29% of proteins upregulated over 4-fold at this exit phase are uncharacterized proteins (Table 1). Among them is Bd1000, which is part of an operon (Bd0997-Bd1000). Interestingly, predicted lipoprotein Bd0998 is also upregulated 3.5-fold at T4h, which could suggest a role for this operon within the exit phase. Further, Bd2744 contains a CHAP domain, which may indicate a peptidoglycan hydrolase or protease function during the exit phase.

### High abundance of protease Bd2269 at exit phase by Western blot confirms quantitative proteome results independently

Comparison of the exit phase with the initial AP revealed protease Bd2269 as one of the most upregulated and highly abundant proteins at the exit phase ($\log_2$(FC) at T3h = 3.47 and at T4h = 3.20, Supplementary Data 3 and Table 1). To validate these data, a Western blot analysis over the whole predatory life cycle was performed using the *B. bacteriovorus* HD100 *bd2269::mCherry* strain, in which the sequence coding for Bd2269 was fused at the C-terminal with mCherry and chromosomally integrated. The Bd2269-mCherry signal pattern confirms the data from the quantitative proteome (Supplementary Fig. 4), first appearing at 2 h, peaking at 3 h, gradually decreasing from 4 to 6 h, before disappearing at 7 h. This semi-quantitative data was confirmed in two independent biological repeats and is consistent with the quantitative proteome results. Since protein abundance and upregulation do not directly indicate function, we performed additional assays to confirm the involvement of Bd2269 in the exit phase.

### Bd2269 mutant exhibits delayed progeny release from prey cells

As a proof-of-concept to demonstrate that the quantitative proteome can reveal specific predator proteins involved in the predatory life cycle, we further investigated the function of Bd2269. To determine the impact of Bd2269 on the predator's exit phase, we tested if *B. bacteriovorus* Δ*bd2269* takes longer to lyse and exit the prey remnants. We performed time-lapse widefield epifluorescence microscopy, measuring the time taken from septation until progeny release from the bdelloplast (exit time). We revealed an increased exit time for *B. bacteriovorus* Δ*bd2269* compared to wildtype HD100 (Fig. 4a). Median exit time was significantly longer in Δ*bd2269* than in wild type (105 min vs. 80 min). Complementation of Δ*bd2269* with pCAT-*bd2269* restored the wild-type phenotype (median exit time = 82.5 min). As a positive control, we used Δ*bd0314*, which lacks a deacetylase-specific predator lysozyme acting on the prey peptidoglycan[39] (median exit time = 110 min). Combining a double deletion of Bd0314 and Bd2269, which targets different components in the exit process, did not reveal a cumulative effect when compared to the single deletion strains (median exit time = 106.3 min). As the deletion mutant progeny cells remained longer in the bdelloplast, these findings imply an important role for Bd2269 in the prey exit process.

### Predator protease Bd2269 causes *E. coli* cell lysis when expressed from within

To assess novel prey cell damaging function of predatory enzymes in the exit phase, we focused on proteases Bd2269, Bd2692, and Bd2321. These predicted secreted proteases are all highly upregulated (Bd2269 9.2-fold, Bd2692 7.3-fold, Bd2321 4.0-fold) during prey exit at 4 h (Fig. 3; Table 1 and Supplementary Data 3), indicating a role in the exit phase. We heterologously expressed them in *E. coli* S17-1 and compared induced vs. repressed conditions to observe potential self-lysis effects. These proteases were expressed with their native, predicted Sec-secretion signal[45], likely transporting them to the *E. coli* periplasm, their site of action. Upon *E. coli* S17-1 cell lysis, internal β-galactosidase is released into the medium with chlorophenol red-β-D-galactopyranoside (CPRG), cleaving off chlorophenol red (CPR), which is detected by fluorescence (Fig. 4b)[32]. Tracking this fluorescence over time under protease-inducing conditions, we observed that Bd2269 lysed *E. coli* cells from within under both normal and high salt conditions (Fig. 4b). High salt conditions were used to increase cellular stress in *E. coli*, potentially diverting resources away from repair mechanisms and thereby enhancing the observable effects of protease-induced damage. Interestingly, under high salt conditions, lysis was also observed in the Bd2269 repressed condition, likely due to a leaky induction system caused by additional salt stress. No lysis was observed for the heterologously expressed Bd2321 and Bd2692 (Fig. 4b). While this assay showed leakiness of the *E. coli* strain expressing the proteases by release of β-galactosidase, it did not assess *E. coli* viability.

**Table 1 | Proteins upregulated (log$_2$(FC) ≥ 2, FDR < 0.05) at exit phase of the predatory life cycle (T4h) grouped according to function**

| Protein name | Function | log$_2$(FC) | Protein name | Function | log$_2$(FC) |
|---|---|---|---|---|---|
| Potential prey cell damaging | | | Replication/Repair | | |
| Bd2269 | Serine protease, subtilase family[a] | 3.20 | Bd0039 | Site-specific DNA inversion stimulation factor | 3.31 |
| Bd2692 | Protease[a] | 2.86 | Bd0512 | Protein RecA | 2.27 |
| Bd3079 | Putative serine proteinase[a] | 2.25 | Bd1582 | Single-stranded DNA-binding protein | 3.20 |
| Bd3507 | Endonuclease I[a] | 2.37 | Bd3382 | Putative DNA-binding protein HU-β | 3.00 |
| Cell motility (gliding[b]) | | | Metabolism | | |
| Bd0414 | Gliding cluster homology | 2.45 | Bd0223 | Putative regulator for granula-associated protein | 2.05 |
| Bd2368 | Gliding cluster homology | 2.83 | Bd0446 | Putative cyclic nucleotide binding | 2.28 |
| Bd2369 | Gliding cluster homology[a] | 2.97 | Bd1874 | Peptidase M48B family[a] | 2.01 |
| Bd2370 | Gliding cluster homology[a] | 2.14 | Bd2354 | THIF family protein | 2.02 |
| Bd2371 | Gliding cluster homology | 3.19 | Bd3486 | 3-deoxy-7-phosphoheptalonate synthase/ chorismate mutase | 2.15 |
| Bd2372 | Gliding cluster homology | 2.57 | Bd3778 | NAD(P)-binding Rossmann-like domain | 2.27 |
| Bd2373 | Gliding cluster homology[a] | 2.82 | | | |
| Bd2374 | Gliding cluster homology[a] | 2.36 | | | |
| Bd2375 | Gliding cluster homology | 2.79 | | | |
| Bd2377 | Gliding cluster homology | 2.31 | Intracellular trafficking and secretion | | |
| | | | Bd2196 | Sec-independent protein translocase protein TatA | 2.14 |
| Division | | | Cell motility (swimming) | | |
| Bd1185 | Activator of cell division | 2.13 | Bd0543 | Flagellar assembly factor FliW | 2.11 |
| Bd1977 | Chromosomal partition | 2.79 | | | |
| Bd2331 | Partition protein, ParA homolog | 2.03 | Function unknown | | |
| Bd3189 | Cell division protein FtsZ | 2.78 | Bd0042 | Uncharacterized protein[a] | 2.15 |
| Bd3190 | Cell division protein FtsA | 2.31 | Bd0278 | Uncharacterized protein[a] | 2.93 |
| Bd3905 | Chromosome partitioning protein | 2.71 | Bd0647 | Uncharacterized protein[a] | 2.21 |
| Bd3906 | Partition protein, ParA homolog | 2.28 | Bd0655 | Uncharacterized protein[a] | 2.64 |
| | | | Bd0774 | Uncharacterized protein[a] | 2.15 |
| ATP hydrolysis | | | Bd1000 | Uncharacterized protein | 2.00 |
| Bd1348 | Putative cation-transporting ATPase[a] | 2.17 | Bd1235 | Uncharacterized protein | 2.71 |
| | | | Bd2361 | Uncharacterized protein | 2.16 |
| DNA synthesis | | | Bd2683 | Uncharacterized protein | 2.32 |
| Bd1983 | Ribonucleoside-diphosphate reductase subunit beta | 2.44 | Bd2715 | BD_b_sandwich domain-containing | 2.66 |
| Bd1984 | Ribonucleoside-diphosphate reductase | 2.68 | Bd2744 | Uncharacterized protein | 2.48 |
| | | | Bd3273 | Putative metalloendopeptidase[a] | 2.39 |
| RNA metabolism | | | Bd3290 | DUF3817 domain-containing | 2.05 |
| Bd2626 | UPF0109 family | 2.21 | Bd3700 | DUF4423 domain-containing | 2.67 |
| | | | Bd3881 | Uncharacterized protein | 2.06 |

[a]Contain secretion signal, predicted with SignalP 6.0[45].
[b]Homology based on Lambert et al. 2010[22].
Protein annotations are provided by either NCBI or eggNOG-mapper v2[33]. Protein name is the old gene locus name.

Therefore, to confirm the results of the β-galactosidase lysis assay, viable cell counts for *E. coli* S17-1 cell survival were performed at the end of the 18-h experiment (Fig. 4c). Induced and repressed conditions for each protease were compared to see their effect on the *E. coli* cell viability. We found that the induction of Bd2269 significantly reduces the number of viable cells in both normal and high salt conditions, supporting the fluorescence measurements (Fig. 4b). Although the lytic capacity of Bd2692 with the β-galactosidase assay under normal salt conditions appears low (Fig. 4b), there is a slight reduction in viable cell counts at moderate significance, showing that Bd2692 damages the *E. coli* cell from within (Fig. 4c). Overall,

these results suggest that protease Bd2269 plays a significant role when the predator exits from the depleted prey and by internal bactericidal activity. Overall, our study demonstrates how our newly generated, highly resolved quantitative predatory proteome over the predatory life cycle can provide important insights to unravel bacterial-damaging enzyme functions.

## Discussion

This study presents a quantitative proteome covering the entire predatory life cycle of *B. bacteriovorus*, capturing the protein expression of 61% of its predatory proteins at each phase in tight succession. Bottom-up proteomics,

**Fig. 4 | Protease Bd2269 is involved in the exit phase of *B. bacteriovorus* and lyses *E. coli* S17-1 from within when heterologously overexpressed.** **a** Exit time comparison of *B. bacteriovorus* HD100 wild type, mutant strains Δ*bd2269*, Δ*bd0314*, and Δ*bd0314* Δ*bd2269*, and complemented strain Δ*bd2269* pCAT-*bd2269* by time-lapse epi-fluorescence microscopy. The exit process was imaged on a 1% agarose pad starting from 3 h post infection of *E. coli* S17-1 pMAL-p2_mCherry prey with a fluorescent cytoplasm and periplasm[78]. Time was measured from progeny septation until first frame of escape from the bdelloplast remnants. Bold black line shows median value with interquartile range. Two-tailed *p* values are shown using the Mann-Whitney test. Two independent biological repeats each with *n* = 30 were evaluated (*n* = 60 per strain). **b** *E. coli* S17-1 damage from proteases Bd2269, Bd2321, and Bd2692 was tested by β-galactosidase activity assay measuring the fluorescence of chlorophenol red under normal (0.171 M NaCl) and high (0.4 M NaCl) salt conditions over 18 h. Bd2269, Bd2321, and Bd2692 were heterologously expressed in *E. coli* S17-1 from a pBAD18-based plasmid under inducing ("_ind") or repressing ("_rep") conditions. "pBAD" stands for the empty vector. *E. coli* S17-1 overexpressing Bd2269 (Bd2269_ind) caused self-lysis and β-galactosidase release over time. Two biological repeats with three technical repeats each were performed and averaged. The plot shows a representative repeat with the second repeat found in Supplementary Fig. 5. **c** Boxplots showing *E. coli* cell viability 18 h after the start of protease induction ("_ind") or repression ("_rep") in different protease expression strains (Bd2269, Bd2321, Bd2692, pBAD [empty vector control]). The median and interquartile range (IQR) were plotted, represented by the box. The whiskers extend to 1.5 times the IQR, and black dots outside of this range are outliers. Two-tailed Welch two-sample *t*-tests compare the mean values between the induced and repressed conditions for each protease (*p*-values are indicated in the graph). Three technical repeats were evaluated across a dilution range of 4.

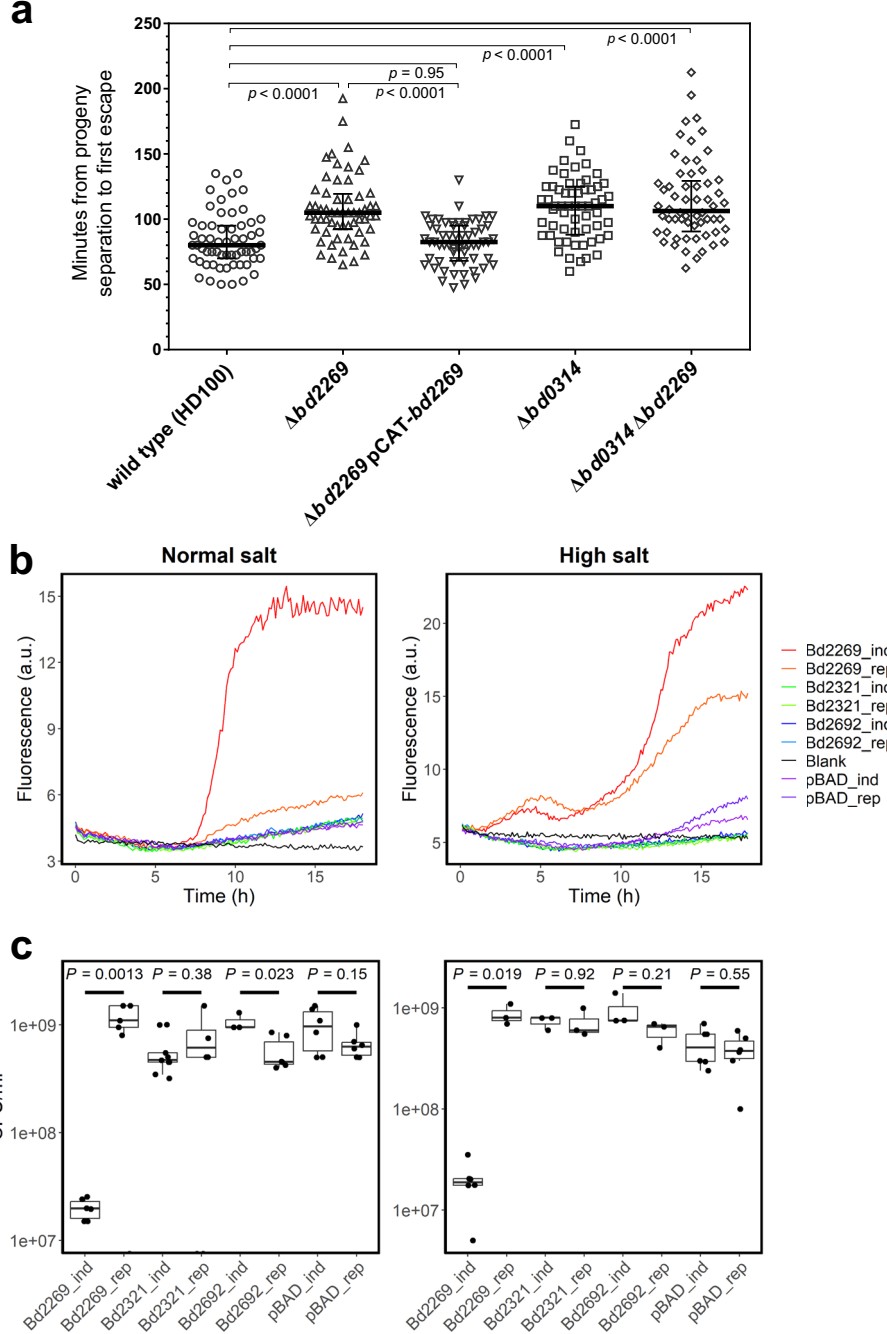

based on enzymatic digestion of proteins to peptides followed by mass spectrometry (MS) analysis combined with relative protein quantification methods, has enabled large-scale protein profiling in a wide range of biological samples. The usage of tandem mass tags (TMT) as an isobaric peptide labeling strategy allows for a highly multiplexed quantitative analysis of multiple biological conditions in a single LC-MS/MS experiment[47,48]. The limitations of that technology are related to the data-independent acquisition mode, which is known to suffer from undersampling issues due to the stochastic selection of precursor signals for identification and quantification. Especially small proteins and low-abundant proteins result in fewer and lower amounts of peptides during enzymatic digestion[49–51]. Furthermore, TMT reporter ion quantification could suffer from ratio compression effects due to co-isolation of co-eluting peptide signals. To partially overcome these limitations, we performed peptide prefractionation to reduce the sample complexity. This decreases co-isolation effects and undersampling issues. Using small precursor isolation windows during MS/

MS acquisition and introducing precursor purity filters during the data analysis helps to reduce the ratio compression effect[49–51]. While we detected no bias towards predicted membrane proteins in the quantitative predator proteome, a larger proportion of small proteins remained unidentified (Supplementary Fig. 2).

However, it would not be expected that all predator proteins would be detected during the predatory life cycle on *E. coli*, since *B. bacteriovorus* may express a fraction of its proteins exclusively under different physiological conditions (e.g., predation on a different prey strain, adaptation to different environmental conditions in AP or during its prey/host-independent life cycle). This quantitative proteome serves as a valuable resource for generating robust hypotheses on predatory enzyme functions that are unknown or predicted in distant organisms. Notably, we demonstrated that the highly upregulated protease Bd2269 plays a critical role in bdelloplast lysis during the predator's exit phase. The insights from this dataset will aid in characterizing the biology of *B. bacteriovorus* and its bacteria-damaging

enzymes, supporting its use as an alternative medical treatment of MDR bacteria[52] or as a preventive measure against plant, animal, and human diseases[53–55].

Our quantitative proteome aligns closely with the cellular biological processes described in the literature for the periplasmic predatory life cycle. While multiple transcriptomic studies on the predatory life cycle of *B. bacteriovorus* have provided a holistic view via varying RNA levels[20,22], our quantitative proteome offers a precise snapshot of the predatory proteome, bringing us closer to understanding the cellular metabolism. Our proteome results differ from the growth phase-specific transcriptome of Karunker and colleagues during this phase[20]. They identified eight genes to represent exclusive growth phase expression (30 min, 1 h, or 3 h), whereas our dataset shows only GroES1 (Bd0097, at T2-4h) and RecA (Bd0512, at T3-4h) with an FC > 2. While semi-quantitative RT-PCRs indicated transcript presence at growth but not AP, our protein data do not support this observation. Looking at the biological context of the mRNA growth-upregulated genes, most serve a housekeeping role (ATP synthesis, ribosomal proteins, electron transport chain). This is likely due to the different lifetimes of protein and RNA in the cells. Differences between the mRNA and protein levels could be explained by post-transcriptional control, including small RNAs and RNA-binding proteins[26,56], or post-translational control, like protein stability. The latter may allow a low-abundant protein to function effectively for an extended period, even though mRNA levels may have begun to drop after translation. Thus, measuring the protein content gives a more accurate account of cellular function compared to mRNA.

Our quantitative dataset provides a rich resource to the research community to advance research on the cellular processes and prey-degradation capabilities of *B. bacteriovorus*. The highly resolved dataset helps to narrow down a subset of uncharacterized predatory proteins at the prey entry and exit phase. To identify predatory proteins secreted into the prey, Tyson et al.[24] analyzed empty bdelloplasts derived from a fraction of failed invasion into the prey periplasm of a MIDAS-adhesin-deficient *B. bacteriovorus* Δbd0875[24]. Their label-free proteomics analysis revealed 155 *B. bacteriovorus* proteins (>2 unique peptides). A comparison with proteins identified in our "Entry cluster A" (Fig. 2b) that contain a signal peptide reveals two proteins of interest: Bd0173 (uncharacterized) and Bd0993 (a predicted secreted glycosidase hydrolase/deacetylase)[57]. Tyson et al.[24] compared the empty bdelloplast proteome for entry-specific proteins with the transcriptome upregulated at 30 min in predation but not HI growth, yielding a refined list of 31 proteins[24] potentially associated with entry into the prey. A comparison of this list with our quantitative proteome revealed that of the 31 identified proteins, we find nine in cluster A (Fig. 2b), four in cluster B, and four were not detected in the quantitative proteome dataset (Supplementary Data 1). Further, two proteins were abundant early, but not associated with a cluster, and twelve proteins showed an expression pattern not upregulated at entry phase. Next to known and anticipated prey peptidoglycan-modifying enzymes (Bd0886, Bd1176, Bd3176, Bd3279)[31,58], all proteins are uncharacterized, while Bd3137 has homology to zinc-binding metalloproteases[59], and Bd1044/Bd1045 are proteins that contain a β-sandwich domain unique to *Bdellovibrio* and related strains. These results show the need for additional future work targeting the mechanism of these uncharacterized proteins that we speculate are involved in prey cell targeting when the predator enters the prey.

Focusing on the prey exit phase, we demonstrate an important functional role of Bd2269 in prey cell lysis. Previously, it was reported that many proteolytic enzymes were secreted by HI strain *B. bacteriovorus* HIB as well as AP *B. bacteriovorus* HD100 when cultured in rich medium, including the three proteases Bd2269, Bd2321, and Bd2692[9,60]. It has been suggested that in the presence of extracellular nutrients, proteases are secreted to release amino acids for the predator's uptake[60]. Furthermore, the application of the HI supernatant or purified proteases Bd2269 and Bd2692 to *Staphylococcus* spp. biofilms resulted in effective dispersal[9,10]. These results indicated that these three proteases are active in two different stages of *B. bacteriovorus* lifestyle: in nutrient uptake and biofilm degradation in AP, and in prey exit. Furthermore, Bd2269 and Bd2692 have been predicted to be highly

expressed based on codon usage[61], further correlating with the protein abundance found in this study. In our study, we test the activity of proteases Bd2269, Bd2321, and Bd2692 in the context of prey cells. We show that the absence of Bd2269 results in delayed exit of prey progeny from the depleted bdelloplast (Fig. 4a). The expression of Bd2269, Bd2321, and Bd2692 heterologously in *E. coli* resulted in a major reduction of *E. coli* cell viability by Bd2269 and to a lesser extent with Bd2692 (Fig. 4b, c). We hypothesize that during the prey exit phase, Bd2269 first degrades proteins into smaller peptides, enabling Bd2692 to break these down further for effective bdelloplast lysis. This is supported by a preference of Bd2692 to target a broad range of smaller peptides and denatured proteins[62]. Our dataset further supports this as Bd2269 is most expressed at T3h, dropping slightly at T4h whereas both Bd2692 and Bd2321 are most highly expressed after Bd2269, at T4h (Supplementary Data 1). Further experiments analyzing Bd2269 in tandem with other proteases could reveal more about how these proteases work together during the exit phase. Since *B. bacteriovorus* Δbd2269, as well as Δbd0314 Δbd2269, can still escape from the prey (Fig. 4a), this indicates that additional factors are involved in *B. bacteriovorus* prey exit. Our quantitative proteome lays the groundwork to narrow down uncharacterized proteins upregulated during this specific phase, enabling future investigations into their mechanisms and functions.

In conclusion, this study offers precise predator protein abundances across the periplasmic predatory life cycle, enhancing our understanding of the predatory bacterium *B. bacteriovorus* and its predatory behavior. Based on the example of the prey-lytic protease Bd2269, we demonstrate how this quantitative predator proteome enables the identification and further functional characterization of proteins involved in key processes of the predatory life cycle. This work lays a foundation for future research and hopes to spur further exploration of *B. bacteriovorus*' complex biology and bacterial prey-damaging capacity. We hope this quantitative proteome will help to uncover *B. bacteriovorus*' potential as a unique antimicrobial solution amidst growing antimicrobial resistance.

## Methods
### Bacterial strains and culturing
*Bdellovibrio bacteriovorus* HD100 strains were maintained in Ca/HEPES buffer (5.94 g/l HEPES free acid, 0.294 g/l calcium chloride dihydrate, pH 7.6) and on solid YPSC agar overlay plates as described previously[63]. Various *E. coli* strains were grown in YT broth at 37 °C, 200 rpm, and supplemented with 25 μg ml$^{-1}$ kanamycin for kanamycin-resistant strains containing plasmids pZMR100[64] or pBAD18[65]. Kanamycin-resistant *B. bacteriovorus* strains were supplemented with 50 μg ml$^{-1}$ kanamycin. A detailed list of primers, plasmids, and strains used in this study can be found in Supplementary Tables 1, 2 and 3, respectively.

### Preparing *B. bacteriovorus* samples over predatory life cycle for quantitative proteomics
Wild-type *B. bacteriovorus* HD100 was grown in Ca/HEPES buffer at 29 °C and 200 rpm for 24 h along with *E. coli* K12 MG1655 prey. The overnight lysate was concentrated by centrifugation at 5000 × *g* for 20 min at 29 °C and resuspended to 1/10 of the original volume in Ca/HEPES. *E. coli* K-12 MG1655 was grown in YT medium at 37 °C and 200 rpm overnight. The *E. coli* culture was concentrated by centrifugation at 5000 × *g* for 10 min at 29 °C, and final optical density 600 nm (OD$_{600nm}$) adjusted to 4.5. Both *B. bacteriovorus* and *E. coli* cultures were incubated at 29 °C, with 200 rpm for 1 h to allow for the stabilization of protein expression. At experiment start, a new lysate was mixed in volume ratio of 1:2:7 containing 10x concentrated *B. bacteriovorus*: washed *E. coli* at OD$_{600nm}$ = 1.0: Ca/HEPES buffer and incubated at 29 °C at 200 rpm. For controls, a respective volume of Ca/HEPES buffer was added to *B. bacteriovorus* and *E. coli*-only cultures and incubated at 29 °C, with 200 rpm for 1 h. At timepoints 15', 30', 45', 1 h, 2 h, 3 h, 4 h, 5 h, 6 h, and 7 h (hence known as sampling timepoints), a 15-ml volume of the lysate was sampled and centrifuged at 5000 × *g* for 20 min at 4 °C. The subsequent supernatant was removed, and the pellet was immediately frozen in liquid nitrogen and stored at −80 °C. After incubation of

the control cultures for 15 min, 15 ml of both *B. bacteriovorus* and *E. coli* only controls were pelleted separately and frozen as above. Five independent biological replicates were performed to properly quantify a 1.5 fold-change for 80% of all the identified proteins with a statistical power of 0.8 and a significance level of 0.05 at growth phase T1h, which was determined in a separate pre-experiment (Supplementary Table 4).

### Processing *B. bacteriovorus* timepoint proteomic samples

The pellets from sampling timepoints were boiled in 100 μl of 2% SDS, and 65 mM dithiothreitol (DTT) in water at 105 °C to denature the proteins. The protein concentrations were measured with the Qubit Broad Range Assay kit (Invitrogen). Forty-five microliters of each sample was processed using SP3, a paramagnetic bead-based method for sample protein cleanup[66]. The proteins were captured on a 1:1 mixture of Sera-Mag carboxylate-modified magnetic beads with a hydrophilic surface (Cytiva Life Sciences, cat. no. 45152105050250, cat. no. 65152105050250). The captured proteins were resuspended in 50 mM tetraethylammonium bromide (TEAB) and digested overnight at room temperature (RT) with 0.012 μg μl$^{-1}$ trypsin (Sequencing Grade, Promega). Following digestion, a pH of ~7–8 was confirmed. Six microliters of the supernatant from each sample were combined to generate a "golden standard" reference, to which samples from different biological repeats can be compared to, next to ensuring comparability between different TMT experiments. The remaining supernatant was dried down in the Speedvac (Thermo Fisher) at 30 °C. The peptide samples were stored at −20 °C until further processing.

### Isobaric labeling of peptides

The peptides were resuspended in 50 mM TEAB buffer at pH 8.3 before addition of 250 μg of a TMTpro-18 plex label (Thermo Fisher) and vortexed briefly. This mixture was incubated for 1 h at RT before quenching with 5% hydroxylamine to a final concentration of 0.3% v/v for 15 min at RT. All samples in the biological repeat were pooled together at an equal ratio. Trifluoroacetic acid (TFA) was added to the pooled sample for a final concentration of 0.5% TFA. The sample was dried in the Speedvac at 30 °C until the final acetonitrile (ACN) concentration was <3% before being cleaned up by C18 solid phase extraction (SPE) (Sep-Pak, Waters) following the manufacturers protocol. Following C18 SPE clean up, the pooled peptide sample was fully dried in the Speedvac at 30 °C and stored at −20 °C until further processing.

### Pre-fractionation and liquid chromatography-tandem mass spectrometry (LC-MS/MS)

High performance liquid chromatographic (HPLC) separation was performed with the Agilent LC1200 HPLC system. The labeled samples were offline pre-fractionated using reverse-phase high-pH chromatography. Peptides were separated on a XBridge Peptide BEH C18 column (130 Å, 3.5 μm, 1 × 100 mm, Waters) using a 60-min linear gradient from 2 to 40% acetonitrile in 9 mM ammonium formate pH 10. A total of 27 fractions were eluted and combined into 9 fractions by pooling fractions 1, 10, 19; fractions 2, 11, 20; etc. The combined fractions were dried by Speedvac and stored at −20 °C until LC-MS/MS measurement.

Mass spectrometry analysis was performed on an Orbitrap Exploris 480 mass spectrometer (Thermo Fisher Scientific) coupled to an M-Class nano-UPLC (Waters). Solvent composition at the two channels was 0.1% formic acid (FA) in water for channel A and 0.1% formic acid in acetonitrile (ACN) for channel B. Column temperature was 50 °C. The dried peptides were dissolved in 3% ACN, 0.1% FA in water. Two microliters of peptides were loaded on a nanoEase MZ Symmetry C18 Trap Column (100 Å, 5 μm, 180 μm × 20 mm, Waters) followed by a nanoEase MZ C18 HSS T3 Column (100 Å, 1.8 μm, 75 μm × 250 mm, Waters). The peptides were eluted at a flow rate of 300 nl min$^{-1}$. After a 3-min initial hold at 5% B, a gradient from 5 to 25% B in 70 min and 25 to 40% B in additional 10 min was applied. The column was cleaned after the run by increasing to 95% B and holding 95% B for 5 min prior to re-establishing and equilibrating the loading condition for 15 min.

Mass spectrometer Orbitrap Exploris 480 was operated in data-dependent mode (DDA) with a maximum cycle time of 3s, using Xcalibur (tune version 4.0 SP1), with spray voltage set to 2.5 kV, funnel RF level at 40%, heated capillary temperature at 275 °C, and Advanced Peak Determination (APD) on. Full-scan MS spectra (350−1500 *m/z*) were acquired at a resolution of 120,000 at 200 *m/z* after accumulation to a target value of 3,000,000 or for a maximum injection time of 45 ms. Precursors with an intensity above 5000 were selected for MS/MS. Ions were isolated using a quadrupole mass filter with 0.7 *m/z* isolation window and fragmented by higher-energy collisional dissociation (HCD) using a normalized collision energy of 32%. HCD spectra were acquired at a resolution of 30,000, and maximum injection time was set to Auto. The automatic gain control (AGC) was set to 100,000 ions. Charge state screening was enabled such that singly, unassigned, and charge states higher than six were rejected. Precursor masses previously selected for MS/MS measurement were excluded from further selection for 20 s, and the exclusion window was set at 10 ppm. The samples were acquired using an internal lock mass calibration on *m/z* 371.1012 and 445.1200. The mass spectrometry proteomics data were handled using the local laboratory information management system (LIMS)[67].

### Quantitative proteome data analysis

The raw data were analyzed in the Proteome Discoverer 2.5 software (Thermo Fisher). Proteins were identified using Sequest HT[68] against a concatenated *B. bacteriovorus* HD100 (UP000008080, *n* = 3583) and *E. coli* K-12 MG1655 (UP000000625, *n* = 4401) proteome, along with a list of common contaminants (*n* = 504). To control the false discovery rate (FDR$_{decoy}$), a target-decoy (reversing protein sequences) approach was used. Within the spectrum selector node, the Sequest HT search parameters were as follows: Fixed modifications: carbamidomethyl, TMTpro on peptide N-terminus; Variable modifications: Oxidation; N-terminal variable modifications: Acetyl, Met-loss, Met-loss+Acetyl; Enzyme: Trypsin; Max. missed cleavages: 2; Peptide charge: 2+ and 3+; Precursor mass tolerance: ±10 ppm; Fragment mass tolerance: ±0.03 Da. The FDR$_{decoy}$ was set to 0.01 for peptide and protein identification. Following the Sequest HT search, Percolator[69] was used for peptide-spectrum match validation and FDR$_{decoy}$ calculations[69]. The resulting raw abundances from the protein level data were imported into R for further analysis (Source file: Quantexp_all_-mod.csv). Identified proteins were filtered for high confidence, and proteins with missing values in any one of the five biological replicates were removed. The raw protein abundances of *B. bacteriovorus* AP and *E. coli* samples were pooled in silico to compare with the later timepoints, where the prey and predator are mixed. The normalizations were done following the protocol of Wilmarth in three steps[70]. First, a "sample loading" (SL) normalization was performed, which uses a scaling factor of the mean of the signal of all TMT channels divided by the total signal in each individual TMT channel. This SL scaling factor is then multiplied by the raw abundances for each protein in each channel. Second, the sample-loading adjusted abundances were normalized by internal reference scaling (IRS)[71]. This was calculated by taking the geometric mean of each protein abundance from the golden standards, dividing this geometric mean by each individual protein abundance, and finally multiplying this result by the sample loading normalization factor. Third, a "trimmed mean of M-values" (TMM) normalization was applied to the IRS normalized data. Following IRS + TMM normalization, the proteins were separated by their respective species and analyzed independently with the prolfqua R-package[72], which included differential expression, linear modeling, and group average estimation (imputation). The protein abundances were filtered for >1, and an additional robust scaling was applied to the data before log$_2$ transformation. For each time point, the normalized abundances from the five TMT experiments (representing five different biological repeats) were averaged, representing the final data (Supplementary Data 1). Comparisons between the AP condition and all other phases of the predatory life cycle were made by calculating the log$_2$ fold change (FC) for each protein. For differential expression, a linear model was performed where the normalized protein abundance was modeled as a function of the

timepoint (Condition) and biological repeats (TMTexp). Two group comparisons against the AP condition were performed using a moderated *t*-test with pooled variance[73]. The p-values were adjusted for multiple testing using Benjamini–Hochberg method, referred to as FDR. This was done for each comparison separately. Imputation of missing values was performed based on group differences of normalized protein abundances. The resulting $\log_2(FC)$ data is in Supplementary Data 3. "eggNOG" annotations were achieved using eggnog-mapper v2[33]. The R script for this data analysis is available at Figshare repository (https://doi.org/10.6084/m9.figshare.29533775).

### Protein length distribution plot of identified *B. bacteriovorus* proteins

To evaluate limitations of the TMT-based quantitative proteome of *B. bacteriovorus*, a protein length distribution plot was generated for membrane and non-membrane proteins in dependency of identification and quantification by mass spectrometry. First, the database file UP000008080_264462_Bdellovibrio_2025_07_07.tsv (available at Figshare, https://doi.org/10.6084/m9.figshare.29533775) was filtered for GO annotation "cellular components" to generate a list of all membrane and non-membrane *B. bacteriovorus* proteins. Second, the proteins identified and quantified in all five TMT experiments (Supplementary Data 1) were used. These two protein lists were compared and separately evaluated for membrane and non-membrane proteins regarding the ratio of identified proteins versus all proteins in the database.

### Clustering of proteins according to expression pattern

An average of the protein abundances from the five biological repeats was calculated from the normalized quantitative proteome. The data was transformed by dividing each condition by the "AP+Ec" value and performing a $\log_2$ transformation such that the "AP+Ec" value is 0. Using this method, the changes from the starting condition can be visualized and clustered based on their pattern throughout the predatory life cycle. Euclidean distance calculations were used for the clustering, and significance thresholds for proteins were filtered for *p*-value < 0.001 and *q*-value < 0.01. A total of 18 clusters were made, and those with very similar cluster profiles were manually combined into a total of ten distinct clusters, nine with changing expression patterns (cluster A-I in Fig. 2b and Supplementary Data 2) and one cluster without notable change containing 839 proteins. Further functional cluster of orthologous genes analysis was performed using eggNOG-mapper v2[33].

### Verification of quantitative proteome by Western blot

*bd2269* was C-terminally fused with *mCherry* and integrated into the *B. bacteriovorus* chromosome by single-crossover homologous recombination. pK18-*bd2269:mCherry* contained the 1-kb region upstream of the 3′ end of the gene with the stop codon removed, along with the mCherry gene downstream. These regions were amplified with Phusion plus (Thermo Fisher) from genomic DNA of *B. bacteriovorus* HD100 and *B. bacteriovorus* HD100 *bd0064:mCherry* and assembled into the suicide vector pK18mobsacB[74] by Gibson assembly[75] (New England Biolabs). The Gibson product was transformed into *E. coli* NEB5α and confirmed by Sanger sequencing before transformation into *E. coli* S17-1. Conjugation was performed between donor *E. coli* S17-1 pK18-*bd2269*:mCherry and *B. bacteriovorus* HD100, with 50 µg ml⁻¹ kanamycin selection as described previously[76,77], and single plaques were isolated and their genotype verified by Sanger sequencing. Cell pellet production with *B. bacteriovorus* pK18-*bd2269*:mCherry preying on *E. coli* K-12 MG1655 pZMR100 was done as described above in "Processing *B. bacteriovorus* timepoint proteomic samples" section, up until protein concentration measurements using the Qubit 4 fluorometer. Protein content was normalized by volume, with concentrations adjusted to ~2 mg ml⁻¹. Equal volumes of each sample were separated by SDS-PAGE and blotted on a nitrocellulose membrane. The membrane was pretreated with SuperSignal Western Blot Enhancer (Thermo Fisher) before hybridization with the anti-mCherry primary

antibody (Invitrogen PA5-34974) and the secondary antibody (Sigma A8275). SuperSignal West Pico PLUS detection reagent (Thermo Fisher) was applied, and proteins were visualized using enhanced chemiluminescence on a LiCor Odyssey system.

### Generation and complementation of protease mutants in *B. bacteriovorus* HD100

Markerless deletion mutants of *B. bacteriovorus* Δ*bd2269* were achieved by taking 1 kb fragments up- and downstream of the gene. As the neighboring downstream gene is transcribed away from *bd2269*, 51 bp was left from the 3′ end of *bd2269* to prevent disruption of the neighboring gene's promoter. The 1 kb regions were assembled into the suicide plasmid pK18*mobsacB*[74] using Gibson Assembly[75] (New England Biolabs), and the resulting pK18-Δ*bd2269* was transformed into *E. coli* NEB5α. After confirmation by Sanger sequencing, the plasmid was further transformed into *E. coli* S17-1 for conjugation into *B. bacteriovorus* HD100 and *B. bacteriovorus* Δ*bd0314* as described previously[39,76,77]. The donor plasmid was cured by multiple rounds of sucrose suicide counter-selection for double-crossover homologous recombination. PCR and Sanger sequencing were performed on single plaque isolates to confirm the deletion of the gene. To complement *B. bacteriovorus* Δ*bd2269*, we generated plasmid pCAT-*bd2269*. *bd2269* was amplified from HD100 genomic DNA, including 196 bp of the upstream region to include the native promoter region. The *bd2269* genomic region was ligated with the pCAT-based plasmid backbone using Gibson assembly[75] (New England Biolabs) and subsequently sequenced and conjugated into *B. bacteriovorus* as described in the "Verification of quantitative proteome by Western blot" section.

### Time-lapse epifluorescence microscopy to determine *B. bacteriovorus* exit duration

Exit time during predation of *B. bacteriovorus* HD100 wild-type, Δ*bd0314*[39], Δ*bd2269*, Δ*bd0314* Δ*bd2269*, or Δ*bd2269* pCAT-*bd2269* on *E. coli* S17-1 pMAL-p2_mCherry[78] prey was examined using time-lapse epifluorescence microscopy. All *B. bacteriovorus* and *E. coli* strains were prepared as described previously[63] and in Section 'Bacterial strains and culturing'. 24-hr lysates of *B. bacteriovorus* microscopy strains were filtered through a 0.45 µm filter (Filtropur S, 0,45, sterile, non-pyrogenic, No. 83.1826, Sarstedt), and 50x concentrated at $3900 \times g$, at 29 °C for 20 min in Ca/HEPES buffer. For Δ*bd2269* pCAT-*bd2269*, an additional wash step was performed to remove the kanamycin. The *E. coli* S17-1 pMAL-p2_mCherry overnight culture was set up containing 10 ml YT, 200 µg ml⁻¹ IPTG and 50 µg ml⁻¹ ampicillin and filtered through a 25 mm sterile prefilter (Millex-AP) before adjustment to $OD_{600}$ = 5.0. The concentrated *B. bacteriovorus* and S17-1 pMAL-p2_mCherry prey were incubated at 29 °C, 200 rpm shaking for 10 min to allow for adaptation to new media. At experiment start, 50 µl of predator and prey were mixed and incubated for 2 h 35 min at 29 °C, 200 rpm. Synchronicity of infection was monitored 30 min after mixing with bright-field light microscopy, checking that at least 90% of the prey cells were rounded based on infection. Two hours 35 min after mixing of predator and prey, 10 µl of the lysate was transferred onto a 1% agarose in Ca/HEPES buffer pad in a µ-petri dish (35 mm, high glass bottom dish, IBIDI, Vitaris AG, Cat. No. 81218-200) for time-lapse microscopy using the Nikon Eclipse Ti2-E widefield system with a 100x objective (CFI Plan Apochromat Lambda 100x Oil, N.A. 1.45, W.D. 0.13 mm, Ph3). The Orca Fusion CoaXpress sCMOS camera and NIS-Elements AR 5.42.01 software (Nikon) were used for image acquisition, taking pictures every 2.5 min for 8 h at 29 °C of nine different fields of view. Phase contrast images were taken with a 300 ms exposure time and light intensity of 40, while mCherry images were taken with an excitation length of 550 nm (LED at 11% intensity) and a 90 ms exposure time. The motorized encoded xy-stage and motorized z-drive, Nikon Perfect Focus System 4, and Okolab Cage incubation box (set to 29 °C and 90% humidity) were used to assure stable incubation conditions and focused imaging during the length of the time-lapse microscopy experiment (starting at ~3 h after mixing predator and prey, recorded for 8 h). Two independent biological replicates, each with 30 events, were

evaluated per strain (total of 60). Exit times were defined by visual inspection from the estimated first frame of visible division of elongated predatory filament into separate progeny cells, to the first frame of visible progeny exit out of the bdelloplast by at least half a progeny cell length. Initial selection of suitable bdelloplasts was supported by NIS-Elements AR software version 6.02.03 (Nikon) and a General Analysis 3 workflow. This workflow segments and numbers potentially suitable bdelloplasts for manual analysis based on fluorescence level, size, circularity, and a suitable exit time within the range of the 8 h imaging.

### Arabinose induced β-galactosidase assay

To construct the strains for the arabinose induced β-galactosidase assay, the *bd2269* and *bd2692* pBAD18-plasmids were constructed by addition of EcoRI and SalI restriction sites. After digestion of pBAD18 with EcoRI and SalI, ligation was performed with T4 DNA ligase (New England Biolabs) at a 1:3 vector to insert ratio and incubated overnight at 16 °C. Alternatively, for the pBAD18-*bd2321* construct, Gibson assembly[75] (New England Biolabs) was performed. Following plasmid ligation, the plasmid was transformed by heat shock into *E. coli* NEB5α and the DNA sequence confirmed by Sanger sequencing. The confirmed plasmids were further transformed into *E. coli* S17-1. The *E. coli* S17-1 pBAD18-*bd2269/2692/2321* strains were grown in LB broth at 37 °C and 200 rpm overnight. 1 ml of the different overnight cultures was washed twice by centrifugation at $5000 \times g$ for 5 min and resuspended in LB broth with normal (0.171 M NaCl) or high (0.4 M NaCl) salt. The pBAD18-*bd2269/2692/2321 E. coli* strains were incubated at 29 °C for 1 h to allow adaptation to the new salt conditions, before adjustment to an $OD_{600nm}$ of 3.0. The induced and repressed conditions were composed of LB medium supplemented with 25 µg ml⁻¹ kanamycin, 30 µM chlorophenol red-b-D-galactopyranoside (CPRG), and 0.2% L-arabinose (for induction) or 0.2% D-glucose (for repression). A black 96-well plate (BRAND, Cat. no 781671) was filled with 125 µl of induced or repressed media and 25 µl of the different pBAD18-*bd2269/2692/2321 E. coli* strains. After inoculating the wells, the plate was placed in a plate reader (Tecan Infinite M200) for 18 h at 37 °C. The fluorescence was measured at $580 \text{ nm}_{Ex}/620 \text{ nm}_{Em}$ every 5 min, with 20 flashes and optimal gain at 100% whilst $OD_{600nm}$ was measured in parallel with 25 flashes. Orbital shaking was performed for 270 s with a 3-mm amplitude in between measurements.

### Statistics and reproducibility

The quantitative proteome of *B. bacteriovorus* was generated with five independent biological replicates per condition. Raw mass spectrometry data were processed using Proteome Discoverer (version 2.5.0) with the SEQUEST HT search engine. Peptide spectrum matches (PSMs) were filtered using a false discovery rate of 1% at the peptide and protein levels. The resulting data were exported from Proteome Discoverer and analyzed in R Studio (version 4.1.1). Data were normalized using the Internal Reference Scaling method[71] and $log_2$ transformed for downstream statistical analysis. The p-values for two-group comparisons against AP were calculated using a moderated *t*-test with pooled variance[73]. Differential expression was assessed using the limma package in R, with *p* values adjusted for multiple testing using the Benjamini–Hochberg method (FDR) for each comparison separately. Proteins with an $-log_{10}(FDR) > 1.3$ and an absolute $log_2$ (fold change) $> 1$ or $< -1$ were considered significantly regulated.

For the arabinose-induced heterologous expression experiments, colony-forming units (CFU/ml) were recorded for both induced and repressed conditions, performed in three technical replicates. CFU/ml between conditions was statistically analyzed using one-way ANOVA followed by pairwise t-tests with Bonferroni correction for multiple comparisons. For targeted comparisons between induced and repressed conditions, Welch's *t*-tests were used to compare the mean values (R Studio, version 4.1.1). The Western blots were generated from biomass from two independent biological repeats. For the analysis of the time-lapse epifluorescence microscopy to monitor the exit duration of different *B. bacteriovorus* strains on *E. coli* S17-1 pMAL-p2_mCherry[78], two independent biological replicates, each with 30 events, were evaluated per strain (n = 60 per strain). Two-

tailed *p*-values were calculated using the Mann–Whitney test using GraphPad Prism® (Version 6.07). Experimental procedures, including sample preparation and LC-MS settings, were standardized and replicated using the same protocols across all repeats to ensure reproducibility.

### Reporting summary

Further information on research design is available in the Nature Portfolio Reporting Summary linked to this article.

## Data availability

All relevant proteomics data have been deposited to the ProteomeXchange Consortium via the PRoteomics IDEntification Database (PRIDE, http://www.ebi.ac.uk/pride)[79] partner repository by EMBL-EBI with the data set identifier PXD056597. All time-lapse microscopy data is available on Bio-Image Archive (https://www.ebi.ac.uk/bioimage-archive/)[80] with Accession number S-BIAD1580 (https://doi.org/10.6019/S-BIAD1580). All further data presented in this study are available in the associated Supplementary Information file, Supplementary Data 1–3 (xls files). Source material, including the source raw data used, is deposited on the Figshare repository (https://doi.org/10.6084/m9.figshare.29533775). The uncropped, original images from the Western Blot in Supplementary Fig. 4 are shown in Supplementary Fig. 6. There are no restrictions on data availability, such as materials transfer agreements. All other data are available from the corresponding author on reasonable request.

## Code availability

The R codes used to process the data for this article are deposited on the Figshare repository with the source material (https://doi.org/10.6084/m9.figshare.29533775).

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

## Acknowledgements

The authors are grateful to Dr. Alexander Ernst (Nikon Europe) for his support with NIS-Elements AR 6.02.03 (64-bit) during time-lapse microscopy data analysis. Microscopy images were obtained with a Nikon Ti2-E microscope maintained by the Center for Microscopy and Image Analysis at University of Zurich. We would like to thank Dr. Ljiljana Mihajlovic (University of Zurich) for help with the cloning for the plasmid for complementation and Prof. Andrew Lovering (University of Birmingham, UK) for very useful comments on the manuscript. The authors would like to thank Prof. Leo Eberl (University of Zurich) for generously hosting the main authors in his laboratory. T.F.L. and S.G.H. were funded by Swiss National Science Foundation Ambizione Fellowship PZ00P3_193401. J.G. and B.R. were funded by the University of Zurich. For the purpose of Open Access, a CC BY public copyright license is applied to any Author Accepted Manuscript (AAM) version arising from this submission.

## Author contributions

T.L. executed all experiments and analysis with the exception of time-lapse microscopy, which D.J. performed. T.L. prepared samples for quantitative proteomics with the support and guidance of B.R. Further, T.L. made statistical analysis of quantitative proteomics data with the help of J.G. All authors contributed to experimental design, which was supervised by S.G.H. T.L. and S.G.H. wrote the manuscript, with additional input on specific sections from D.J., B.R., and J.G. All authors contributed to the article and approved the final manuscript submitted.

## Competing interests

The authors declare no competing interests.
