## [Transparent Peer Review file · Communications Biology]

Quantitative proteome of bacterial periplasmic predation by *Bdellovibrio bacteriovorus* reveals a prey-lytic protease

Corresponding Author: Dr Simona Huwiler

This manuscript has been previously submitted at another journal. This document only contains information relating to versions considered at Communications Biology.

Version 0:

Reviewer comments:

Reviewer #1

(Remarks to the Author)

The study of Lai et al., employs advanced quantitative proteomics to profile the entire predatory life cycle of *B. bacteriovorus*, identifying 2,195 proteins and uncovering nine distinct expression clusters. Notably, the protease Bd2269 was found to be highly upregulated during the predator's exit phase, suggesting its role in damaging the prey during progeny release. Functional validation through gene knockout and heterologous expression experiments supports this hypothesis, highlighting Bd2269 as a critical factor in the predator's life cycle.

This work aligns with recent studies that emphasize the potential of *B. bacteriovorus* as a biocontrol agent against multidrug-resistant Gram-negative pathogens. The identification of Bd2269 adds to the growing body of knowledge regarding the enzymatic arsenal of this predator and its applications in antimicrobial therapy and biotechnology.

Minor points:

Although the data normalization and the use of limma for statistical analysis are explained, the main text lacks details on the significance criteria (p-value, FDR) and whether corrections for multiple comparisons were applied in the key contrasts. Could the authors comment on the limitations of using TMT-based quantitative proteomics for detecting low-abundance proteins? Specifically, how might this have affected the identification of rare or weakly expressed predatory enzymes in *B. bacteriovorus*?

Figure 3: authors should include the color legend of the genes in each volcano plot.

Line 577: doi of the R script for the data analysis is missing

Reviewer #2

(Remarks to the Author)

The authors have provided a very extensive data set in a quantitative analysis of the proteome of the type strain of *Bdellovibrio bacteriovorus* over the entire predatory life cycle. This information is undoubtedly an important resource for all researchers in this field of study. Most importantly, it should facilitate research into the search for novel antibacterial enzymes that could be useful in the treatment of antimicrobial resistant pathogens.

General Comments

The title of the manuscript doesn't quite reflect the balance of the data. The study mainly presents "the first quantitative proteome covering the complete predatory life cycle" of *Bdellovibrio bacteriovorus*. The fact that the proteome reveals an interesting protease predicted to be involved in the prey exit process is interesting indeed, but the experiments to analyze the functional role of Bd2269 seem to read as an 'add on' and would be better suited for a separate manuscript. Such a study would illustrate the use of the proteomic database for analysis of specific stages in the predatory life cycle.

I recommend that the name of the bacterium under study should be in the title of the article, not just in the 'keywords' section.

I'm not so sure that the term used in the title and throughout the manuscript – "prey damaging" – is appropriate. What do the

authors mean by “prey damaging”? In my experience microbiologists don’t usually refer to a ‘damage’ to a cell. There is the concept of bacteriostatic and bacteriocidal antimicrobial agents, but the discussion does not usually involve damage to a cell, rather inactivation – i.e. growth or no growth. Lytic enzymes are definitely needed for the new progeny cells to escape the bdelloplast. However, bdelloplasts are not cells, but a structure (or ‘growth container’). If the authors wish to continue to use the term “prey damaging capacity” then it needs to be explained better.

Specific Comments

1. Ln 22: “the bacterial predator” is not needed as this is already stated in Ln 18.
2. Ln 37-38: I think Ref. 4 should be added after “living antibiotic” because that reference essentially defines the term. Then use Ref. 5 after “immunogenicity”.
3. Ln 40-41: should also add here that *B. bacteriovorus* inhibits the formation of biofilms of *S. aureus* (ie not just ‘reduces’ biofilms).
4. Ln 43: add “as it” before “is genetically-tractable ... “
5. Ln 44-46: the concept here of the Gram-negative cell envelope (prey or not) is not correct. In a Gram-negative bacterium the cell ‘wall’ consists of the peptidoglycan, periplasm and outer membrane. The cell ‘envelope’ includes the plasma membrane. Therefore, the predator invades through the outer membrane and peptidoglycan, not the “outer leaflet of the prey”. Also, the predator does not feed on the contents of the periplasm as a nutrient source. The cytoplasmic contents are the main source of nutrients.
See also Ln 54-55. As explained above, the outer membrane and cell wall are not separate entities.
6. Ln 53: enzymatic functions that could serve as a potential source of new antimicrobial drugs.
7. Ln 62: please add “culture” to supernatant. Otherwise, the source of the supernatant is not clear.
8. Ln 78: “AP” needs to be defined here, not in Ln 102.
9. Ln 105-106: further to my earlier point: what do you mean by “internal cell damage”?
10. Ln 119: Is “the model organism” really necessary to describe *E. coli*?
11. Fig. 1a: correct spelling of *Bdellovibrio*. Also, the flagellum of this predator has a damped waveform and should be illustrated as such.
12. Ln 170: what does “extremely regulated” mean?
13. Ln 203: contact to the prey cell is via the pilus structure, not the unassembled pilin protein.
14. Ln 311: “break open” is not a good scientific term.
15. Ln 312-313: “until progeny release from the bdelloplast”, not “leaving”
16. Ln 324 forward: please explain why high salt conditions (salt stress) was tested with *E. coli*.
17. Ln 387: a bdelloplast cannot be “dead”. It is not a cell. It is a structure, a container for growth of progeny cells inside.
18. Ln 434: I don’t think that after 60+ years of research on BALOs that *B. bacteriovorus* is “underexplored”. Research is going along just fine.
19. List of References: With regard to the title of publications – some titles have all terms with the first letter in capitals; other titles only have the first word capitalized.

Version 1:

Reviewer comments:

Reviewer #1

(Remarks to the Author)

The authors have incorporated methodological criteria for statistical significance and multiple testing correction, explained the limitations of TMT proteomic technology, improved Figure 3 for easier interpretation, and have adjusted key biological terminology and contexts to better integrate the proteomic and functional findings.

Reviewer #2

(Remarks to the Author)

Response by authors continues with the three asterisks at the start.
Reply by reviewer follows.

*** We agree with the reviewer, that in principle the manuscript is composed of two parts, which could potentially be published in two separate articles: the first quantitative proteome of *Bdellovibrio bacteriovorus* over the whole predatory life cycle, and the protease Bd2269, identified in the first part, but mechanistically shown to have an impact on the release of *B. bacteriovorus* from the Bdelloplast by knock out studies and heterologous expression studies showing the internal damaging potential in *E. coli*. The authors joined these two parts together, to aim for a higher impact of the manuscript, and to show how the systematic quantitative proteome data set can directly translate into the discovery of experimentally proven enzymes involved in this predator prey interaction with bacterial lysis potential.

The authors have provided a good explanation for the format of the article and I accept their decision to keep the two parts of the article. The revisions in the revised manuscript to improve the rationale for including the data on Bd2269 are well done.

***The authors did not include B. bacteriovorus in the title initially, as to the best of our knowledge this is the first quantitative proteome of bacterial periplasmic predation overall.

Then even more important to indicate which bacterium was studied.

*** The authors see the argument on the difficulty of using the term “prey damaging”.

The authors have made good revisions to clarify the enzymatic activity of Bd2269. Thank you for using the term ‘bactericidal’.

Specific Comments

Ln 119: Is “the model organism” really necessary to describe E. coli?

***Yes, we would like to keep “the model organism” in this location of the text to make clear to the reader, why the E.coli laboratory strain K-12 MG1655 is used, rather than a E.coli clinical strain (which might be considered regarding the AMR resistance crisis and in the clinical context).

The rationale is good. So I recommend that in the new line 127 you add “the laboratory strain” to the text. E.g. “...after mixing predator with the model organism Escherichia coli K-12 MG1655 (a laboratory strain) as prey ... “

Fig. 1a: correct spelling of Bdellovibrio. Also, the flagellum of this predator has a damped waveform and should be illustrated as such.

*** Thank you for spotting the typo. This was corrected. The Fig. 1a was adjusted to show a predator with a flagellum with damped waveform.

The artwork on the flagellum was well done.

Author response to reviewers and the editor for manuscript "Quantitative proteome of bacterial periplasmic predation reveals a prey damaging protease"

***Replies by authors are marked with asterisks at the start and in blue.

Reviewers' comments:

Reviewer #1 (Remarks to the Author):

The study of Lai et al., employs advanced quantitative proteomics to profile the entire predatory life cycle of *B. bacteriovorus*, identifying 2,195 proteins and uncovering nine distinct expression clusters. Notably, the protease Bd2269 was found to be highly upregulated during the predator's exit phase, suggesting its role in damaging the prey during progeny release. Functional validation through gene knockout and heterologous expression experiments supports this hypothesis, highlighting Bd2269 as a critical factor in the predator's life cycle.

This work aligns with recent studies that emphasize the potential of *B. bacteriovorus* as a biocontrol agent against multidrug-resistant Gram-negative pathogens. The identification of Bd2269 adds to the growing body of knowledge regarding the enzymatic arsenal of this predator and its applications in antimicrobial therapy and biotechnology.

*** The authors would like to thank the reviewer for his/her valuable feedback and time invested reviewing our work to improve it, especially with regards to the proteomic work.

Minor points:

Although the data normalization and the use of limma for statistical analysis are explained, the main text lacks details on the significance criteria (p-value, FDR) and whether corrections for multiple comparisons were applied in the key contrasts.

***Thank you for this valuable feedback. We now implemented significance criteria in the main text (see current lines 192-194). These thresholds are additionally indicated in the lines in the volcano plots in Fig.3. Multiple testing correction was applied to all the tests. The FDR refers to an adjusted p-value using the Benjamini-Hochberg method in the main text, data analysis, Supplementary Table 3 and Figure 3. For each contrast (comparison of two different stages of the life cycle protein levels/two group comparison) the Benjamini-Hochberg correction was applied separately. This was added as an additional sentence in the Methods and Materials part on current lines 614/615. Further, to avoid confusion of the different FDR in the Methods and Materials part, we introduced an FDR_{decoy} which is used in a different context to control the false discovery rate by a target decoy for peptide and protein identification (lines 581, 586, 588). Additionally, an additional section on "Statistics and Reproducibility" was inserted into the Methods and Materials part (lines 748 - 773).

Could the authors comment on the limitations of using TMT-based quantitative proteomics for detecting low-abundance proteins? Specifically, how might this have affected the identification of rare or weakly expressed predatory enzymes in *B. bacteriovorus*?

*** Thank you very much for this comment. Bottom-up proteomics methodologies that use data-dependent acquisition methods have well-known limitations due to the stochastic selection of precursor ions. One such limitation is the approximately 3–5 order of magnitude dynamic range of mass spectrometry, which is insufficient to cover the biological dynamic range of protein concentrations (>10 orders of magnitude). Consequently, peptides from highly abundant proteins often dominate the ion population, resulting in the suppression of less abundant peptides and hindering their detection. Additionally, low-abundant and small proteins (under 100 amino acids) yield fewer and lower amounts of peptides during enzymatic digestion.

To address these limitations, reducing sample complexity at the protein or peptide level, as well as applying enrichment techniques, could improve the sensitivity of the analysis. At the mass spectrometry level, a data-independent acquisition technique could overcome the limitations of stochastic precursor signal selection, or targeted methods such as parallel reaction monitoring (PRM) could increase the sensitivity of MS analysis. We combined peptide prefractionation to reduce sample complexity and isobaric peptide tags to increase the signal-to-noise ratio for peptides, partially overcoming the limitations of the proteomics experiment.

We now point out the limitations of TMT-based quantitative proteomics and low-abundance proteins as well as weakly expressed proteins, next to membrane proteins (as requested by the editor) in the revised manuscript at the beginning of the results section (new lines 133-134) as well as the discussion (lines 375-396). An additional supplementary figure 2 was implemented in the revised manuscript, showing in detail the limitations of the generated TMT-based quantitative proteomics towards detection of smaller proteins, whereas there is no bias that membrane proteins from *B. bacteriovorus* were underrepresented. To generate supplementary figure 2 an additional description was inserted in the methods and materials part (lines 621-631).

Figure 3: authors should include the color legend of the genes in each volcano plot.

*** Thank you for this comment. To improve the Figure overview, we implemented an overall legend with color code and grouping of the different genes in the new Figure 3. The top left volcano plot of the comparison of 15' versus "attack phase" was changed slightly by moving the protein labels aside, so that all dots on the volcano plots can be seen.

Line 577: doi of the R script for the data analysis is missing

***The R scripts can be found in the publicly available doi on FigShare upon publication. As the openly available, published data on figshare can not be changed anymore, the authors provide here a different private link to all reviewers (next to the new cover letter), to access the R script and all source data for this manuscript immediately:

<https://figshare.com/s/e721f066211785a48899>. The data available via this private link will be made publicly available during the proof reading stage. The doi for public access upon publication was implemented in current lines 619, 626 and 788 (currently not working).

Reviewer #2 (Remarks to the Author):

The authors have provided a very extensive data set in a quantitative analysis of the proteome of the type strain of *Bdellovibrio bacteriovorus* over the entire predatory life cycle. This information is undoubtedly an important resource for all researchers in this field of study. Most importantly, it should facilitate research into the search for novel antibacterial enzymes that could be useful in the treatment of antimicrobial resistant pathogens.

*** The author would like to thank the reviewer for his/her estimation of our work. We are grateful to receive his/her valuable feedback and reflections, while thanking his/her for the time invested reviewing our work to improve it, particularly regarding the predatory bacterial aspects.

General Comments

The title of the manuscript doesn't quite reflect the balance of the data. The study mainly presents "the first quantitative proteome covering the complete predatory life cycle" of *Bdellovibrio bacteriovorus*. The fact that the proteome reveals an interesting protease predicted to be involved in the prey exit process is interesting indeed, but the experiments to analyze the functional role of Bd2269 seem to read as an 'add on' and would be better suited for a separate manuscript. Such a study would illustrate the use of the proteomic database for analysis of specific stages in the predatory life cycle.

*** We agree with the reviewer, that in principle the manuscript is composed of two parts, which could potentially be published in two separate articles: the first quantitative proteome of *Bdellovibrio bacteriovorus* over the whole predatory life cycle, and the protease Bd2269, identified in the first part, but mechanistically shown to have an impact on the release of *B. bacteriovorus* from the Bdelloplast by knock out studies and heterologous expression studies showing the internal damaging potential in *E.coli*. The authors joined these two parts together, to aim for a higher impact of the manuscript, and to show how the systematic quantitative proteome data set can directly translate into the discovery of experimentally proven enzymes involved in this predator prey interaction with bacterial lysis potential. We improved the revised manuscript by connecting the two parts better as advised by the editor (abstract lines 26-32, introduction lines 111-117, results lines 320-322, discussion lines 473-476).

Finally, based on the current academic situation of some authors, we decided to keep these two parts together as the manuscript was assessed in peer review containing both parts. While separating these two parts could overall lead to an additional publication and summed-up higher impact for the authors, this would lead to an estimated delay in publication of the peer-reviewed article by at least a year, hindering next steps in the academic careers.

I recommend that the name of the bacterium under study should be in the title of the article, not just in the 'keywords' section.

*** We agree to change the title to: "Quantitative proteome of bacterial periplasmic predation by *Bdellovibrio bacteriovorus* reveals a prey-lytic protease". The authors did not include *B. bacteriovorus* in the title initially, as to the best of our knowledge this is the first quantitative proteome of bacterial periplasmic predation overall.

I'm not so sure that the term used in the title and throughout the manuscript – “prey damaging” – is appropriate. What do the authors mean by “prey damaging”? In my experience microbiologists don't usually refer to a ‘damage’ to a cell. There is the concept of bacteriostatic and bacteriocidal antimicrobial agents, but the discussion does not usually involve damage to a cell, rather inactivation – i.e. growth or no growth. Lytic enzymes are definitely needed for the new progeny cells to escape the bdelloplast. However, bdelloplasts are not cells, but a structure (or ‘growth container’). If the authors wish to continue to use the term “prey damaging capacity” then it needs to be explained better.

*** The authors see the argument on the difficulty of using the term “prey damaging”.

We improved the manuscript to explain the “prey damaging” effect more clearly by replacing it with prey lysis in case of heterologous expression of Bd2269 in *E. coli* (lines 30, 113, 348, 879, 893, in Supplementary information line 83). To increase clarity, the authors now use “bacteriocidal” when referring to Fig.4c (line 366). While the Bd2269 naturally will act on the bdelloplast, it has shown to be lytic and bacteriocidal to *E.coli* (prey) in Fig.4c. when expressed in the “not natural” way from within *E.coli* itself. We would like to keep the lysis aspect and in some sentences the term “damage” remains, as the authors would like to show using Bd2269 as an example, how predatory enzymes have the capacity to damage/lyse other bacterial cells to inspire the field to look for further, new antimicrobial acting predator enzymes.

Specific Comments

1. Ln 22: “the bacterial predator” is not needed as this is already stated in Ln 18.

***Agreed, it was removed (new line 23).

2. Ln 37-38: I think Ref. 4 should be added after “living antibiotic” because that reference essentially defines the term. Then use Ref. 5 after “immunogenicity”.

*** We cited the References more precisely (new line 41). The first time the term ‘living antibiotics’ was coined by Prof. Liz Sockett as to the best of our knowledge in the *Nat Rev Microbiol.* publication of 2004 (page 672, first paragraph on “Therapeutic potential”, <https://doi.org/10.1038/nrmicro959>). Hence, we swapped the references around and added the one of Sockett and Lambert directly after the term ‘living antibiotic’.

3. Ln 40-41: should also add here that *B. bacteriovorus* inhibits the formation of biofilms of *S. aureus* (ie not just ‘reduces’ biofilms).

*** Thank you for this input. This was included in the manuscript on line 44-45.

4. Ln 43: add “as it” before “is genetically-tractable ... “

***This was implemented (new line 47).

5. Ln 44-46: the concept here of the Gram-negative cell envelope (prey or not) is not correct. In a Gram-negative bacterium the cell ‘wall’ consists of the peptidoglycan, periplasm and outer

membrane. The cell 'envelope' includes the plasma membrane. Therefore, the predator invades through the outer membrane and peptidoglycan, not the "outer leaflet of the prey". Also, the predator does not feed on the contents of the periplasm as a nutrient source. The cytoplasmic contents are the main source of nutrients.

***Thank you for this comment, which was implemented accordingly (new line 48-50).

See also Ln 54-55. As explained above, the outer membrane and cell wall are not separate entities.

***Thank you for pointing this out. We have changed it accordingly (new line 61).

6. Ln 53: enzymatic functions that could serve as a potential source of new antimicrobial drugs.

***This improvement was implemented (new line 58).

7. Ln 62: please add "culture" to supernatant. Otherwise, the source of the supernatant is not clear.

***Agreed, this is more precise (new line 68).

8. Ln 78: "AP" needs to be defined here, not in Ln 102.

***Thank you for spotting this. It was changed accordingly (new lines 85 and 109).

9. Ln 105-106: further to my earlier point: what do you mean by "internal cell damage"?

***By "internal cell damage" we mean that Bd2269, if expressed from an inducible and tightly controlled plasmid within *E.coli* (heterologous expression), causes *E.coli* damage from within upon induction. In Fig.4b we show that *E.coli* internal beta-galactosidase is leaking out depending on induction of Bd2269 within *E.coli* leading to an increased fluorescence of reporter chlorophenol due to conversion by beta-galactosidase. Based on this "leaking out of internal beta-galactosidase" in the presence of Bd2269 we infer "internal cell damage". Further, in Fig. 4c we show that the *E.coli* cell viability is reduced about 100-fold if Bd2269 is internally expressed. To increase clarity in the manuscript text we changed the text accordingly (removal or change of "internally") in lines 113, 336-337, 348-349, 366, 878-879, 893 and lines 809, SI line 83-84)

10. Ln 119: Is "the model organism" really necessary to describe *E. coli*?

***Yes, we would like to keep "the model organism" in this location of the text to make clear to the reader, why the *E.coli* laboratory strain K-12 MG1655 is used, rather than a *E.coli* clinical strain (which might be considered regarding the AMR resistance crisis and in the clinical context). Further, it will help the reader to understand why *E.coli* was taken as a "model prey" to determine this first full quantitative proteome over the whole predatory life cycle, rather than an other Gram-negative prey.

11. Fig. 1a: correct spelling of *Bdellovibrio*. Also, the flagellum of this predator has a damped waveform and should be illustrated as such.

*** Thank you for spotting the typo. This was corrected. The Fig. 1a was adjusted to show a predator with a flagellum with damped waveform.

12. Ln 170: what does “extremely regulated” mean?

***Thank you for pointing out that this phrasing is unclear. In principle the authors mean that most up or down regulated proteins can be spotted best and easiest in the volcano plot and more detailed information is available at once when comparing two conditions with a volcano plot. However, as in general a more detailed view is possible using the volcano plot, also for the less regulated proteins, we now generalised and changed the sentence accordingly. (at new lines 179-180).

13. Ln 203: contact to the prey cell is via the pilus structure, not the unassembled pilin protein.

***Thank you for this remark. We removed this sentence (now lines 214-215 in new manuscript).

14. Ln 311: “break open” is not a good scientific term.

*** We replaced “break open” with “lyse” (new line 320).

15. Ln 312-313: “until progeny release from the bdelloplast”, not “leaving”

*** This was adjusted (new line 323).

16. Ln 324 forward: please explain why high salt conditions (salt stress) was tested with *E. coli*.

*** Thank you for input that this is not well explained. We improved the manuscript in lines 349-351 to incorporate this. Higher salt conditions were used to increase the stress on *E. coli* overall, next to the heterologous expression of the predator proteases. The authors were hoping that some additional stress on the cell might increase the chances to detect a damaging effect of the different predator proteases, as the *E. coli* cell is using resources to already counteract the salt stress and might have less capacity to repair a smaller internal damage by predator proteases. However, the study showed that there was only an effect with protease Bd2269 but not the other proteases, irrespective of the salt (stress) conditions (Fig. 4bc).

17. Ln 387: a bdelloplast cannot be “dead”. It is not a cell. It is a structure, a container for growth of progeny cells inside.

*** Thank you for your comment. We removed “dead” in this context to prevent confusion (new line 425). It was initially inserted to point out to the reader that the bdelloplast is a non-living structure.

18. Ln 434: I don’t think that after 60+ years of research on BALOs that *B. bacteriovorus* is “underexplored”. Research is going along just fine.

***The authors want to apologize for using this not well matching expression. We simply wanted to express, that *B. bacteriovorus* is generally less explored than many other bacteria used as model organisms. The authors removed “underexplored” (new line 472).

19. List of References: With regard to the title of publications – some titles have all terms with the first letter in capitals; other titles only have the first word capitalized.

***This was adjusted in the newly submitted manuscript.

The following final changes were made to the manuscript as requested by the editor and reviewer2:

All line numbers refer to the final manuscript without track change mode.

Reviewer2:

The rationale is good. So I recommend that in the new line 127 you add “the laboratory strain” to the text. E.g. “...after mixing predator with the model organism *Escherichia coli* K-12 MG1655 (a laboratory strain) as prey ... “

** This was implemented in line 121.

Editor:

** Line 192: at the end of the subtitle “..form” was added.

Comment:

As the Supplementary Data legends are very long and extensive, they were kept in the SI PDF document, but also short legends provided in the Editorial requests table for the online platform.

**Further changes in connection with finalizing the manuscript:
(according to the final revisions instructions)**

** Abstract text (lines 18-29): Results part changed to present tense.

**Various locations: Supplementary Tabel 1-3 was renamed to Supplementary Data 1-3, the Supplementary Tables 4-6 were renumbered to Supplementary Tables 1-3.

**Line 777-783: Rephrasing of what data is found where. The Source raw data file will be available on the Figshare repository.

** Line 501: In the brackets newly, “Supplementary Table 4” is mentioned, providing this new table for the estimation of the biological replicates in the Supplementary Information PDF. Further this sentence (line 500-503) was rephrased slightly, and a detected minor error detect (as the calculation was only run for 70% and 80% of proteins identified the 75% was corrected to 80%).

****Data availability statement:**

Addition of a sentence that Suppl. Fig. 6 shows uncropped image of Suppl. Fig. 4. A similar sentence was added to the end of legend of Suppl. 4 including a statement that the second biological repeat data can be found in the figshare folder. New Suppl. Fig. 6 was added to Suppl. Information file.

Addition of: All other data are available from the corresponding author on reasonable request.

**** Table 1: bold titles in the table were made non-bold, but underlined.**

**** Supplementary Information file:**

Adjusted page number on first page for index

-New Supplementary Figure 6

-New Supplementary Table 4

****Acknowledgement**

The Acknowledgements section was slightly adjusted. E.g. The thank you for donated strains was removed.